# Identification of multiple *Acinetobacter baumannii* protein antigens as targets for potential immunotherapies using a novel protein microarray screening approach

Samantha Palethorpe[1], Giuseppe Ercoli[1], Elisa Ramos-Sevillano[1], Gathoni Kamuyu[1], Joe Campo[2], Samuel Willcocks[3], Rie Nakajima[4], Philip Felgner[4], Brendan Wren[3], Ganjana Lertmemongkolchai[5], Richard Stabler[3], Jeremy Brown[1]*

1 UCL Respiratory, University College London, London, United Kingdom, 2 Antigen Discovery Inc., Irvine, California, United States of America, 3 Department of Infection Biology, London School of Hygiene and Tropical Medicine, London, United Kingdom, 4 Vaccine Research and Development Center, Department of Physiology and Biophysics, University of California Irvine, Irvine, California, United States of America, 5 Cellular and Molecular Immunology Unit, Centre for Research and Development of Medical Diagnostic Laboratories (CMDL), Faculty of Associated Medical Sciences, Khon Kaen University, Khon Kaen, Thailand

* jeremy.brown@ucl.ac.uk

## Abstract

The World Health Organisation has identified *Acinetobacter baumannii* as a critical priority antimicrobial resistant (AMR) pathogen for which new therapeutics are needed. Despite this, currently there are no antibody or vaccine candidates in advanced clinical development for *A. baumannii*. To help address this, we designed a protein microarray approach to identify multiple *A. baumannii* protein antigens for further investigation as potential targets for vaccination or an antibody therapy. An 868-protein microarray was constructed containing mainly highly conserved *A. baumannii* proteins, and was enriched for those predicted to be surface localised and for which the corresponding gene is highly expressed during culture in *ex vivo* human serum. Probing the protein microarray with sera obtained from mice after non-lethal infection with multiple different *A. baumannii* strains identified IgG responses to 66 proteins. Four proteins (three previously poorly described outer membrane proteins and BamA, a known protective vaccine antigen selected as a positive control) were selected for further investigation. Polyclonal rabbit IgG to all four protein antigens recognised multiple clinical AMR *A. baumannii* strains, and for selected strains promoted opsonisation with IgG and complement, improved neutrophil phagocytosis, and increased membrane attack complex formation. Passive immunisation with polyclonal IgG to each antigen partially protected mice against *A. baumannii* sepsis, and a combination of polyclonal to two antigens completely protected against *A. baumannii* murine sepsis. Repeating passive immunisation experiments in mice depleted of complement, neutrophils or tissue macrophages demonstrated protection

**Data availability statement:** Raw RNAseq data were uploaded to the European Nucleotide Archive (ENA) and the individual data file accession numbers are as follows: ERR8982504, ERR8982505, ERR8982506, ERR8982507, ERR8982508, ERR8982509, ERR8982510, ERR8982511, ERR8982512, ERR8982513, ERR8982514, ERR8982515, ERR8982516, ERR8982517, ERR8982518, ERR8982519, ERR8982520, ERR8982521 (DOI: 10.3389/fimmu.2022.853690). All remaining data supporting the findings of this study are available within the manuscript and its Supplementary Data files.

**Funding:** The work was undertaken at UCLH/UCL who receive funding from the Department of Health's NIHR Biomedical Research Centre's funding scheme, and was supported by the Medical Research Council grant MR/S004394/1 and the BactiVac network grant BVNCP6-03. SP received salary support from MR/S004394/1 and BVNCP6-03, GK from MR/S004394/1, and GE and ERS from a Wellcome Investigator award 221803/Z/20/Z. The funders had no role in study design, data collection and analysis, decision to publish, or preparation of the manuscript.

**Competing interests:** I have read the journal's policy and the authors of this manuscript have the following competing interests: The authors have filed a patent application based on the results reported in this study (patent application number: 2416072.3).

against systemic infection was dependent on complement and neutrophils but not macrophages. Overall, the data demonstrate that our protein microarray is a novel approach that can rapidly identify multiple new protein antigens as potential antibody targets for preventing or treating AMR bacterial infections.

## Author summary

Antimicrobial-resistant (AMR) bacterial infections are predicted to cause 10 million deaths per year by 2050. One important AMR bacterium is *Acinetobacter baumannii*, which typically causes multidrug-resistant infections in hospitalised patients and is a particularly common problem in some lower- and middle- income countries. Vaccines or monoclonal antibody therapies could reduce the burden of infections caused by AMR *A. baumannii*, but need suitable target antigens that are conserved among clinical strains of *A. baumannii* and promote bacterial killing by the immune system. Here, we use a novel protein microarray technique to identify multiple novel target protein antigens that are present in the majority of clinical *A. baumannii* isolates and are expressed during infection. Further testing showed that polyclonal rabbit antibodies raised against selected antigenic proteins promoted immune recognition of clinical *A. baumannii* strains and increased clearance of multidrug-resistant *A. baumannii* infections in mice. Our study provides a novel approach for identifying protein antigen targets for bacterial pathogens that could lead to improved prevention or treatment of AMR infections.

## Introduction

The World Health Organisation (WHO) has identified antimicrobial resistance (AMR) as a major threat to human health, responsible for an estimated 4.95 million deaths worldwide in 2019 [1]. One of the top priorities for the development of new antimicrobials is the Gram-negative bacterium *Acinetobacter baumannii* [2], which causes a range of infections including pneumonia, septicaemia, urinary tract infections, and skin/soft tissue infections [3–5]. *A. baumannii* was a relatively rare cause of disease in the 1970s [3], but since then the incidence of *A. baumannii* infections has increased rapidly and is now estimated to cause over 300,000 deaths/year globally [1]. *A. baumannii* infections are especially problematic in Southeast Asia, causing 25,000 deaths/year in Thailand alone [4]. Around 80% of Thai clinical *A. baumannii* isolates are multidrug-resistant (MDR) [6] and resistance rates to colistin, the antibiotic of last resort, are 11.2% [7]. Consequently, untreatable infections are occurring, contributing to the reported mortality rates of up to 70% for *A. baumannii* infections [4,8].

One approach to reducing the morbidity and mortality caused by bacterial AMR infections is immunotherapy, either through (i) active vaccination or (ii)

administration of monoclonal antibodies (mAbs) to prevent infections in high-risk patients (e.g., passive vaccination of intensive care patients), or (iii) as an additional therapy in combination with antibiotics in people with proven infection [9]. The potential of antibody therapies is shown by the successful use of convalescent sera to treat some bacterial infections in the pre-antibiotic era, the present routine use of mAbs to treat or prevent viral infections [10–12], and the existing mAb therapies for *Bacillus anthracis* (raxibacumab, obiltoxaximab) and *Clostridium difficile* (bezlotoxumab) [13]. The extensive range of routine clinical mAb therapies targeting human antigens as well as pathogens means novel mAbs for clinical use can be developed rapidly. Furthermore, resistance to an antibody therapy is much less likely to develop than to novel antibiotics [14].

The potential advantages of active vaccination or mAb prophylaxis or treatment means these approaches are being widely considered for multiple AMR pathogens [13,15,16]. However, currently there are no active vaccine or mAb candidates in advanced clinical development for MDR *A. baumannii* infections [2]. This is partially due to several significant difficulties in identifying suitable target antigens for immunotherapies. The obvious antigen target for many bacterial pathogens is the extracellular polysaccharide capsule, which is both surface exposed and present in high quantities. For example, all existing *S. pneumoniae* vaccines use the capsule as their target antigen, and in mice, vaccination with the *A. baumannii* capsule protects against subsequent infection [16–18]. However, the genetic locus encoding the *A. baumannii* capsule shows considerable variation between strains, with over 100 described capsular genotypes (termed KL types), most of which will vary in their biochemical structure and hence will also have different antibody specificities [19,20]. Multiple different antibodies would be required to cover the large number of KL types found in clinical *A. baumannii* isolates (e.g., 30 + KL types present in 191 Thai *A. baumannii* clinical isolates, with the commonest KL type found in only 16% of isolates [20]) making it difficult to use capsular antigen as a target for vaccines or mAb therapies. Alternatively, immunotherapies could target subcapsular protein antigens that are conserved amongst infecting *A. baumannii* strains. However, similar to many other bacterial pathogens, *A. baumannii* has an extensive pangenome with a high proportion of surface proteins that are either absent or show substantial allelic variation between strains. Furthermore, the capsule partially inhibits access of the immune response to subcapsular antigens [21,22]. Despite this, several *A. baumannii* subcapsular protein antigen candidates have been identified as targets for novel antibody therapies or active vaccines for MDR *A. baumannii* [23], including outer membrane proteins (OMPs) (e.g., Oxa-23 [16], OmpA [24], BamA [25–27], Omp22 [28], Omp34 [29], and iron-regulated outer membrane proteins (IROMPs) [30,31]), and the outer membrane nuclease NucAB [32]. These data indicate that protein antigens could be effective targets for vaccines and mAbs for preventing or treating AMR *A. baumannii* infections [16]. Identifying multiple additional potential protein antigen targets would increase the chances of successful development of an immunotherapy and would also increase the available target antigen options for multivalent approaches which could improve vaccine or mAb efficacy.

Here, we have used a novel approach based on an 868-protein antigen microarray to rapidly identify multiple antigenic *A. baumannii* proteins which could be potential therapeutic targets for immunotherapy approaches. Additionally, we investigated the efficacy of polyclonal rabbit antibody to four of these antigens at promoting immune recognition of clinical MDR *A. baumannii* strains *in vitro* and for protecting mice against *A. baumannii* infection.

## Results

### Selection of proteins for the *A. baumannii* protein microarray

Using genome data from 220 *A. baumannii* strains including 191 clinical Thai isolates [20], an 868 *A. baumannii* protein microarray was constructed for immunogenicity studies to identify potential protein antigens suitable as targets for immunotherapeutics. To increase their suitability as targets for immunotherapy, the proteins selected for inclusion in the microarray were enriched for those with: (i) a high degree of conservation between *A. baumannii* strains amongst the clinical isolates described by Loraine *et al*. [20]; (ii) a predicted surface location; and/or (iii) higher levels of expression of

PLOS Pathogens

the corresponding gene during culture *in vitro* in human serum as a marker for likely expression during human infection. For proteins included in the microarray, the corresponding gene is present in a mean of 215.4 (97.9%, range 51.8-100%) of the 220 *A. baumannii* genomes when analysed using a 95% DNA conservation level, and 89.9% of genes are present in 90 + % of these strains [20] (**Fig 1A**). The predicted localisation of proteins selected for the microarray is the bacterial envelope (periplasm, inner or outer membrane) for 580 proteins (66.8%), extracellular for 22 (2.5%), cytoplasmic for 155 (17.9%), and unknown for 111 (12.8%) (**Fig 1B**). To identify genes that were preferentially expressed in serum, RNAseq was performed using three Thai clinical strains (AB1, AB1615-09, and NPRC-AB20) incubated in either *ex vivo* serum or broth culture. Gene expression in serum correlated closely between the three strains (**Fig 1C and 1D**), with 124 genes (14.3%) showing increased expression for all three strains in serum compared to broth (**Fig 1E**). Proteins selected for the microarray were enriched for those for which the corresponding gene showed increased expression in serum (**Fig 1F**). Overall, for proteins included in the microarray, 431, 370 and 189 of the corresponding genes had increased expression in serum compared to culture in broth for strain AB1, AB1615-09, and NPRC-AB20, respectively. The conservation, predicted localisation, and relative expression of the corresponding gene in serum are listed for each protein included in the protein microarray in **S1 Table**.

## Identification of immunogenic *A. baumannii* proteins

To identify *A. baumannii* proteins capable of inducing immune responses in mice, the protein microarray was probed with sera obtained from mice one month after non-lethal infection with *A. baumannii*. Seven groups of mice were infected with different *A. baumannii* regimens using Thai clinical isolates. The regimens included either three exposures to a single strain of *A. baumannii* (single strain sera) for five different strains, or three exposures using a different strain each time for two strain combinations that only partially overlapped with the other regimens (multi-strain sera) (**Fig 2A and S2 Table**) [33]. The same high but non-lethal challenge dose of $1 \times 10^6$ CFU was used for each *A. baumannii* strain; the combination of a high inoculum dose, multiple different clinical strains, and repeated inoculations (known to increase the number of antigens recognised [33]) were used to ensure antibody responses were elicited for a broad range of potential antigen targets. Whole cell ELISAs against both an encapsulated (AB5075 WT) and an unencapsulated (AB5075$^{wza}$) *A. baumannii* strain demonstrated significant IgG recognition for sera from most infected mice groups, indicating significant IgG was present to *A. baumannii* protein antigens (**Fig 2B**). Unexpectedly, there were higher ELISA titres in serum from mice vaccinated with the mixed strain regimens compared to a single strain. When used to probe the microarray, each infecting strain/ combination of strains showed significant IgG responses to between 11 and 35 protein antigens, with shared recognition of many antigens between vaccination regimens (e.g., ABUW_2730 and ABUW_1741) (**Fig 2C and 2D and S1 Table**). Inoculation with the mixed *A. baumannii* strain regimens resulted in an almost four-fold increase in the mean number of antigens recognised by IgG compared to sera from mice infected with a single strain (**Fig 2D**), with principal component analysis showing clear separation in the immune response to *A. baumannii* proteins in sera obtained from the two multi-strain vaccination combinations (**Fig 2E**). The MFI for IgG binding to individual antigens varied both within and between vaccination regimens (**Fig 2C**). Overall, a total of 66 protein antigens induced IgG in at least one mouse (**Table 1**). **Fig 2F** shows the mean MFI for IgG binding for all serum samples for the top 20 recognised antigens across all vaccination groups.

## Selection of immunogenic *A. baumannii* proteins for further investigation

Four antigenic proteins were selected for further investigation for their potential as targets for protective immunotherapy (**Table 2**). Ag1 (ABUW_0166, a predicted porin), Ag5 (ABUW_2887, a predicted lipoprotein), and Ag7 (ABUW_3834, a predicted peptidyl-prolyl isomerase) were selected as (a) they are all predicted OMPs, (b) they have not previously been described as a potential target antigen for an antibody therapy or vaccine, and (c) their corresponding genes are present in 99% or more of 842 *A. baumannii* genomes on the NCBI website, representing strains isolated from multiple countries

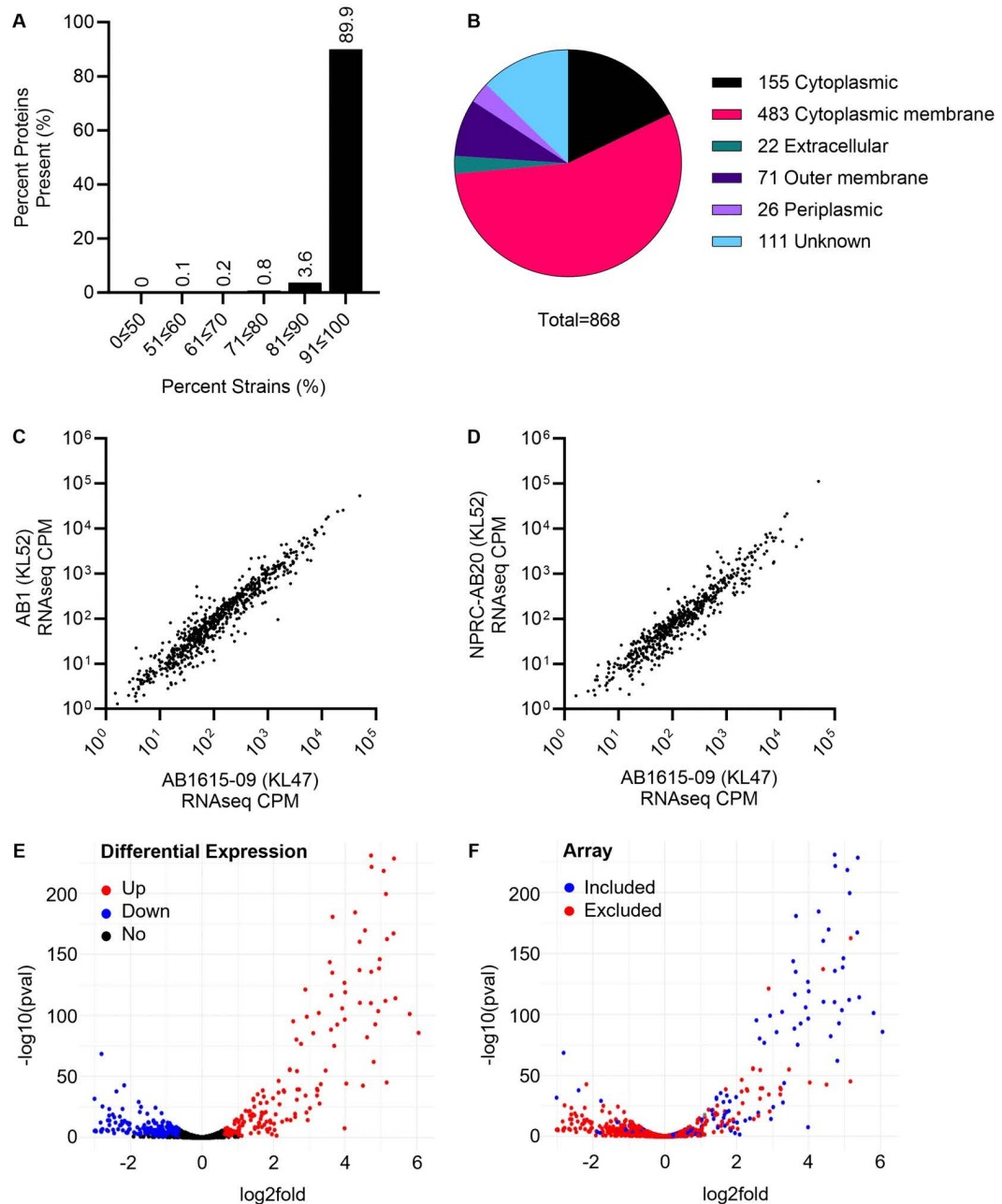

**Fig 1. Selection of proteins for *A. baumannii* protein microarray. (A)** Percent of selected microarray proteins present in 220 clinical *A. baumannii* isolates described by Loraine *et al*. [20]. **(B)** Cellular localisation of selected microarray proteins predicted using PSORT (v3.0) and the *A. baumannii* strain AB1615-09 protein sequences. **(C)** and **(D)** Correlations of normalised transcript copy number (counts per million, CPM) in human sera for genes for the proteins included in the microarray proteins for strains AB1615-09 versus AB1 **(C)**, and AB1615-09 versus NPRC-AB20 **(D)**. KL; K locus capsule type. **(E)** Volcano plot of combined RNAseq data using Salmon quantification and Sleuth analysis; combined highly conserved, differentially expressed genes (from strains AB1615-09, AB1, and NPRC-AB20) with a $\log_2$ fold expression > 0.6 and a *p* value < 0.05 (red = upregulated in human sera compared to LB broth, blue = downregulated, black = no change). **(F)** Volcano plot from panel **(E)** recoloured to show the CDSs included in the protein microarray (blue = included, red = excluded).

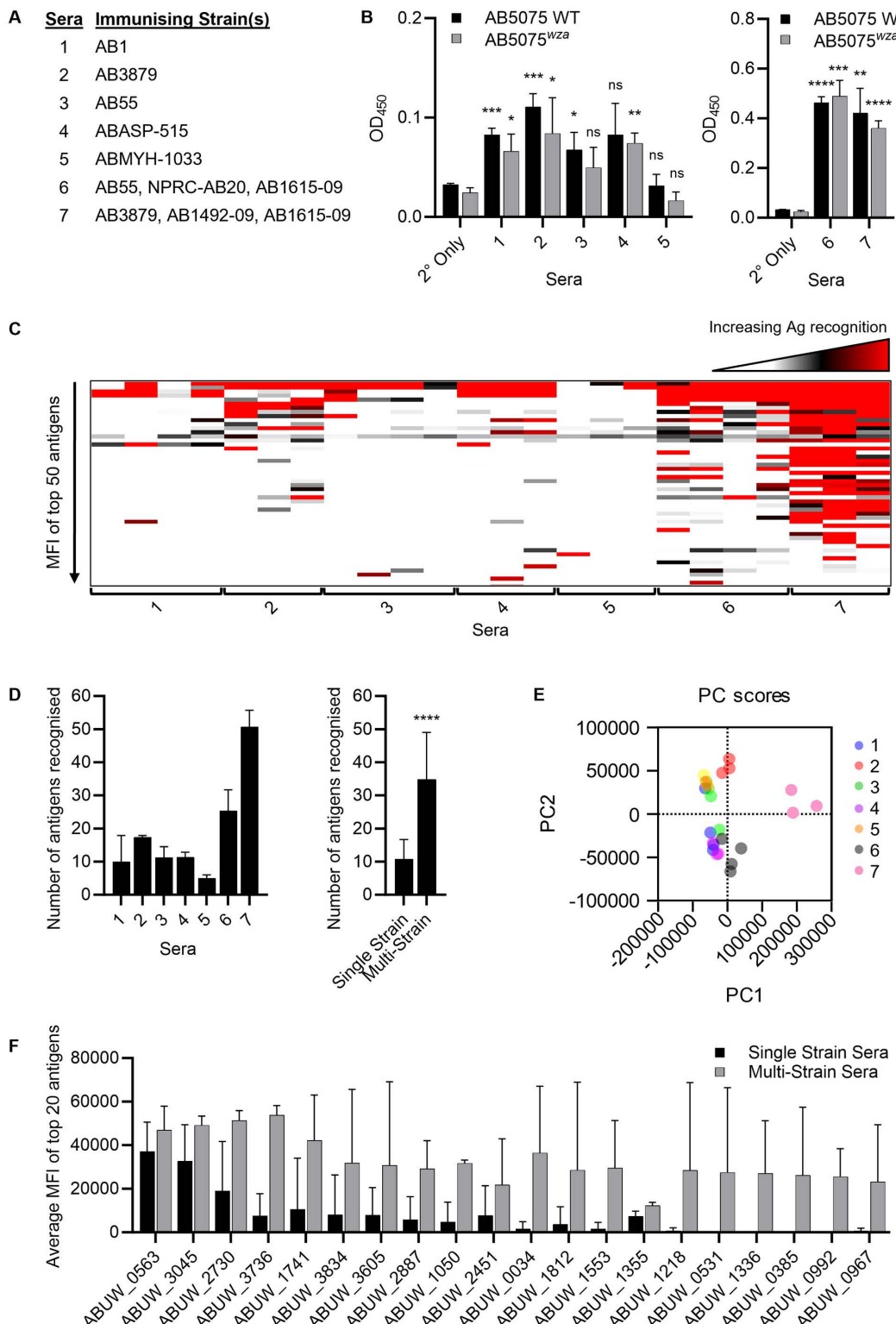

**Fig 2. Identification of antigenic *A. baumannii* proteins. (A)** Immunisation groups listing strains used for mice receiving three sub-lethal infections of either the same *A. baumannii* strain (single strain sera, 1-5) or three different *A. baumannii* strains (multi-strain sera, 6 & 7) [33] used to obtain sera for whole cell ELISAs and to probe the *A. baumannii* protein microarray. **(B)** Mean ODs (error bars = SDs) for whole cell ELISAs for the *A. baumannii* strain

AB5075 WT and its unencapsulated *wza* mutant probed with 1:100 single strain sera (left) or multi-strain sera (right) from immunised mice (sera number refers to panel **A**) (2° only = no sera, secondary antibody only control). **(C)** Results of probing the microarray with mouse sera from groups described in panel **A,** represented as a heat map of IgG MFIs for each mouse for the top 50 antigens. **(D)** Mean (error bars = SDs) number of proteins promoting a significant IgG response when assessed by probing the microarray with sera described in panel **A**; left panel data presented separately for each immunising strain(s), right panel pooled results for single versus multi-strain immunisation groups. **(E)** Principal component analysis of microarray data showing separation of the results for each immunisation groups. **(F)** Mean MFI for the top 20 antigens identified by probing the protein microarray with mouse sera from the 7 immunisation groups. Black columns, combined data for single strain immunised mice, grey columns combined data for multi-strain immunised mice. Data were analysed using unpaired Student's t-tests (*$p < 0.05$, **$p < 0.01$, ***$p < 0.001$, ****$p < 0.0001$, ns; not significant) in panels **(B)** (versus PBS control mice) and **(D)** (single versus multi-strain results).

and continents, including 127 different Pasteur Multilocus Sequence Types (M405 strains from the dominant clinical lineage ST2) and 36 isolates with no currently defined MLSTs (**S3 Table**). The fourth protein selected was BamA (Ag11, ABUW_1741), an outer membrane β-barrel assembly protein that is essential for outer membrane biogenesis [34], which has previously been described as stimulating protective immunity when used as a vaccine and hence was selected as a positive control [25–27].

### IgG to the selected target antigens recognises clinical *A. baumannii* strains

Polyclonal rabbit antibodies were raised against each of the selected purified protein antigens (**Table 2**) for use in *in vitro* and *in vivo* experiments. In Western blots using denaturing gels against purified protein or whole cell lysates of the AB5075 Wild Type (WT) strain (KL25) and two Thai clinical strains representing different KL serotypes and STs (AB3879 (KL10) and AB98 (KL47), **S2 Table**), the polyclonal rabbit antibodies recognised proteins corresponding to the predicted sizes for each antigen (**Fig 3A to 3D**). As expected, Ag1, Ag5, and Ag7 were not detected in the corresponding AB5075 transposon mutant strains (A5075$^{ag1}$, A5075$^{ag5}$, and A5075$^{ag7}$, respectively) (**Fig 3A to 3C**). An Ag11 mutant was not tested as *bamA* is an essential gene [34]. There was non-specific binding to additional proteins in bacterial lysates for α-Ag1 (**Fig 3A**), and these responses may affect the ELISA titres for this antigen (presented below).

Fluorescent microscopy was used to demonstrate whether polyclonal IgG bound to the surface of the encapsulated laboratory *A. baumannii* strain AB5075 (KL25), and the Thai clinical strain AB3879 (KL10). Since the *A. baumannii* capsule often blocks access to subcapsular protein antigens [21,22], we also included the AB5075 isogenic unencapsulated mutant strain, AB5075$^{wza}$ (**Fig 3E to 3G**). Incubation with IgG to all four antigens resulted in significant fluorescence to the unencapsulated strain, although the pattern of the binding varied significantly between antigen targets and between *A. baumannii* strains. Probing strain AB5075$^{wza}$ with α-Ag5 and α-Ag7 resulted in focal dots of fluorescence on the bacterial surface, whereas α-Ag1 and α-Ag11 resulted in a uniform layer of fluorescence around the bacteria (**Fig 3E**). As expected, there was reduced IgG binding to the encapsulated AB5075 WT strain probed with α-Ag5 and α-Ag7 with only the occasional foci of fluorescence visible. In contrast, probing the encapsulated *A. baumannii* AB5075 WT strain with α-Ag1 and α-Ag11 still resulted in a uniform layer of fluorescent surrounding the bacterial cell (**Fig 3F**). Probing the encapsulated clinical AB3879 strain with α-Ag5 also produced a speckled pattern of fluorescence similar to that seen for strain AB5075$^{wza}$. However, in contrast to the results with the AB5075 strains, probing AB3879 (KL10) with α-Ag7 IgG resulted in uniform binding around the bacteria (**Fig 3G**).

Whole cell ELISAs against all three strains also showed significant recognition by all four polyclonal antibodies, with increased binding to the unencapsulated strain as expected (**Fig 3H to 3J**). The ability of purified rabbit polyclonal IgG to Ag1, Ag5, Ag7, and Ag11 to promote immune recognition of clinical *A. baumannii* strains was also investigated using whole cell ELISAs against eight additional clinical *A. baumannii* isolates chosen to represent several KL serotypes and STs (**Fig 4** and **S2 Table**) [20,33,35]. There were significant IgG ELISA titres to all the clinical *A. baumannii* strains when incubated with each polyclonal IgG. The strength of recognition between strains and antigenic target varied slightly,

**Table 1. List of 66 proteins inducing significant IgG responses in at least one mouse when measured by probing the *A. baumannii* protein microarray with sera obtained from groups of mice after non-lethal infection with seven different *A. baumannii* regimens (as described in the text and Fig 2) [33].**

| AB5075-UW annotation | Predicted function | Amino Acid Conservation (%)[a] | Mean IgG response[β] | RNAseq serum (CPM) for AB1615-09[δ] | RNAseq broth vs serum q value |
|---|---|---|---|---|---|
| ABUW_2730 | OmpA family protein | 84 | 28214 | 1501 | 0.0903 |
| ABUW_2724 | protein tolA | 100 | 3761 | 5350 | 0.1835 |
| ABUW_2554 | putative hemolysin | 98 | 4720 | 1293 | 0.0097 |
| ABUW_2451 | TetR family regulatory protein | 99 | 11743 | 146 | 0.0004 |
| ABUW_2325 | Putative transcriptional regulator | 98 | 344 | 15 | 0.0004 |
| ABUW_2259 | lipid A export permease/ATP-binding protein MsbA | 100 | 1650 | 301 | 0.4989 |
| ABUW_2222 | ABC-type dipeptide/oligopeptide/nickel transport systems permease component | 100 | 2265 | 29 | 0.1522 |
| ABUW_2096 | Acetoacetyl-CoA transferase, alpha subunit | 100 | 301 | 1958 | 0.0004 |
| ABUW_1530 | metal ion ABC transporter substrate-binding protein/surface antigen | 97 | 2690 | 441 | 0.0579 |
| ABUW_1553 | cyoA | 100 | 9606 | 2220 | 0.0603 |
| ABUW_1572 | Putative 3-hydroxyacyl-CoA dehydrogenase | 100 | 5113 | 57 | 0.0004 |
| ABUW_1648 | Putative transporter with mechanosensitive ion channel | 99 | 626 | 130 | 0.0007 |
| ABUW_1741 | putative outer membrane protein **(antigen 11)** | 100 | 19570 | 441 | 0.0325 |
| ABUW_1772 | Phosphotransferase system, fructose-specific EI/HPr/EIIA components | 100 | 1181 | 56 | 0.1191 |
| ABUW_0385 | toluene tolerance efflux transporter (ABC super-family, PerI-bind) | 100 | 7489 | 0 | 0.0000 |
| ABUW_0383 | toluene tolerance efflux transporter (ABC super-family, ATP-bind) | 100 | 2227 | 1616 | 0.0245 |
| ABUW_0359 | Putative signal peptide-containing protein | 100 | 1018 | 297 | 0.0212 |
| ABUW_0603 | signal peptide | 99 | 511 | 10847 | 0.0004 |
| ABUW_0676 | nolF | 100 | 3941 | 53 | 0.8868 |
| ABUW_0698 | bifunctional sulfite reductase flavoprotein alpha-component/iron-uptake factor | 100 | 716 | 215 | 0.0004 |
| ABUW_0826 | putative DcaP-like protein | 81 | 274 | 1184 | 0.1898 |
| ABUW_0842 | outer membrane protein | 100 | 2904 | 0 | 0.0000 |
| ABUW_0870 | sdhA | 100 | 6323 | 571 | 0.7209 |
| ABUW_3424 | ExbD protein | 98 | 470 | 0 | 0.0000 |
| ABUW_3426 | TonB protein | 98 | 6840 | 896 | 0.0004 |
| ABUW_3505 | magnesium and cobalt efflux protein CorC | 100 | 418 | 0 | 0.0000 |
| ABUW_1063 | hypothetical protein | 92 | 1784 | 0 | 0.0000 |
| ABUW_1050 | glyceraldehyde-3-phosphate dehydrogenase | 100 | 12443 | 537 | 0.0628 |
| ABUW_1030 | lepB | 100 | 463 | 222 | 0.1401 |
| ABUW_0995 | tolR | 100 | 972 | 351 | 0.2461 |
| ABUW_0994 | protein TolA | 100 | 6945 | 60 | 0.8250 |
| ABUW_0993 | tolB | 100 | 720 | 214 | 0.0404 |
| ABUW_0992 | pal | 100 | 7299 | 784 | 0.7322 |

*(Continued)*

**PLOS Pathogens**

**Table 1.** (Continued)

| AB5075-UW annotation | Predicted function | Amino Acid Conservation (%)[α] | Mean IgG response[β] | RNAseq serum (CPM) for AB1615-09[δ] | RNAseq broth vs serum q value |
|---|---|---|---|---|---|
| ABUW_0967 | membrane-fusion protein | 100 | 7052 | 100 | 0.9897 |
| ABUW_0922 | multidrug efflux pump | 100 | 472 | 105 | 0.0004 |
| ABUW_3153 | multidrug resistance efflux pump | 100 | 5182 | 42 | 0.2288 |
| ABUW_2984 | signal peptide | 81 | 5730 | 771 | 0.0004 |
| ABUW_2910 | efflux ABC transporter permease | 100 | 308 | 34 | 0.1497 |
| ABUW_2898 | putative signal peptide-containing protein | 52 | 1269 | 529 | 0.1981 |
| ABUW_2887 | putative lipoprotein **(antigen 5)** | 100 | 12487 | 789 | 0.0004 |
| ABUW_3834 | FKBP-type peptidyl-prolyl cis-trans isomerase **(antigen 7)** | 89 | 14893 | 407 | 0.5693 |
| ABUW_3756 | Protein of unknown function (DUF445) | 100 | 1720 | 106 | 0.0265 |
| ABUW_3736 | ATP synthase F0F1 subunit B | 100 | 20705 | 1339 | 0.0004 |
| ABUW_0034 | Putative RND family drug transporter | 100 | 11493 | 42 | 0.0004 |
| ABUW_0035 | Acr family drug resistance transporter | 100 | 1008 | 43 | 0.0004 |
| ABUW_1812 | hypothetical protein | 100 | 10748 | 268 | 0.0230 |
| ABUW_1355 | hemin uptake protein HemP | 100 | 8812 | 50476 | 0.0004 |
| ABUW_1336 | Putative RND family drug transporter | 100 | 7740 | 4.0 | 0.9515 |
| ABUW_1131 | nlpD | 100 | 6480 | 321 | 0.5548 |
| ABUW_1120 | subtilisin-like serine protease | 96 | 2989 | 95 | 0.5965 |
| ABUW_0184 | acetate permease | 100 | 5627 | 394 | 0.7762 |
| ABUW_0166 | omp25 **(antigen 1)** | 100 | 4897 | 8055 | 0.2980 |
| ABUW_1218 | HtrA-like serine protease | 100 | 8600 | 259 | 0.2480 |
| ABUW_3045 | putative outer membrane protein | 100 | 37487 | 9954 | 0.9744 |
| ABUW_3020 | RND family multidrug resistance secretion protein | 100 | 2582 | 46 | 0.0004 |
| ABUW_3644 | permease | 100 | 1329 | 403 | 0.0694 |
| ABUW_3605 | FHA domain protein | 100 | 14447 | 326 | 0.6457 |
| ABUW_3903 | putative inner membrane protein translocase component YidC | 98 | 6421 | 347 | 0.0046 |
| ABUW_3885 | purK | 100 | 351 | 161 | 0.0152 |
| ABUW_3375 | hypothetical protein | 100 | 1401 | 0 | 0.0000 |
| ABUW_3364 | outer membrane protein | 100 | 774 | 58 | 0.0004 |
| ABUW_3362 | membrane-fusion protein | 100 | 6027 | 357 | 0.0004 |
| ABUW_3087 | putative competence protein | 100 | 6190 | 410 | 0.0408 |
| ABUW_0505 | OmpA/MotB protein | 100 | 1098 | 380 | 0.0494 |
| ABUW_0531 | secD | 100 | 7866 | 230 | 0.0004 |
| ABUW_0563 | beta-lactamase OXA-23 | 70 | 39922 | 4436 | 0.0079 |

[α]Degree of conservation of the predicted protein product amongst 220 clinical *A. baumannii* isolates [20].

[β]MFI obtained by probing the protein microarray with sera from mice inoculated with both multi-strain combinations of *A. baumannii* strains [33].

[δ]Average transcript copy number in serum for strain AB1615-09 was 553 CPM (counts per million)

sometimes independently of KL type (e.g., compare α-Ag11 results for the KL47 strains AB15 and AB1615-09, **Fig 4A and 4B**). Overall, α-Ag7 showed the highest IgG ELISA titre for the highest number of strains (7 out of 8) and α-Ag11 had the lowest titres for the highest number of strains (7 out of 8).

**Table 2. Summary of the data for the *A. baumannii* proteins selected as potential targets for an antibody therapy or active vaccine. Full protein microarray and RNAseq data for each antigen are shown in S1 Table.**

| Ag# | AB5075-UW annotation | Predicted mW (kDa) | Predicted function | Gene Conservation (%)[a] | Mean IgG response (MFI)[β] | RNAseq serum (CPM)[δ] | RNAseq broth vs serum q value |
|---|---|---|---|---|---|---|---|
| Ag1 | ABUW_0166 | 27.7 | Porin | 99.5 (838) | 4898 | 8055 | 0.2980 |
| Ag5 | ABUW_2887 | 17.5 | Lipoprotein | 99.6 (839) | 11470 | 789 | 0.0004 |
| Ag7 | ABUW_3834 | 26.0 | Peptidyl-prolyl isomerase | 99.0 (834) | 14893 | 407 | 0.5693 |
| Ag11 | ABUW_1741 | 93.8 | BamA | 83.5 (703) | 19570 | 441 | 0.0325 |

[a]The proportion as a percentage (number in brackets = actual number of strains) of 842 strains with genomes available on the NCBI website (see S3 Table) containing a gene with 95% identity at the DNA level to the corresponding AB5075 gene.

[β]Mean Fluorescent Intensity obtained by probing the protein microarray with sera from mice inoculated with all single strain/multi-strain combinations of *A. baumannii* isolates [33].

[δ]Average transcript copy number in serum for strain AB1615-09 was 553 CPM (counts per million).

## Antibody to selected antigens promoted immune recognition and opsonophagocytosis of clinical *A. baumannii* isolates

To assess the potential immune consequences of antibody recognition of the selected antigens, opsonisation with IgG and complement components C3b/iC3b, neutrophil phagocytosis, and complement membrane attack complex formation (MAC, C5b-8/C5b-9) were assessed using established flow cytometry assays [33,35] (**Fig 5A to 5H**). Two relevant multidrug-resistant clinical *A. baumannii* strains (AB3879 (KL10) and AB98 (KL47)) were selected for these assays to represent different KL serotypes and STs (**S2 Table**). As previously reported [33,35], after incubation in normal human serum (NHS) *A. baumannii* strains AB3879 and AB98 varied in their susceptibility to C3b/iC3b opsonisation, neutrophil phagocytosis, and membrane attack complex (MAC) formation (**Fig 5C to 5H**). In general, there was stronger promotion of immune recognition by antibody to the target antigens for the AB98 strain compared to the AB3879 strain. With the exception of α-Ag7 for the AB3879 strain, antibody to all the selected antigens supported significant levels of IgG opsonisation of both AB3879 and AB98 (**Fig 5A and 5B**). Antibody to Ag1 and Ag5 increased C3b/iC3b deposition against AB3879, and antibody to all four antigens increased C3b/iC3b deposition on the AB98 strain (**Fig 5C and 5D**). Complement-dependent neutrophil phagocytosis of both strain AB3879 and AB98 was improved by α-Ag5 and α-Ag7 but not α-Ag1 and α-Ag11 (**Fig 5E and 5F**). C5b-8/C5b-9 formation was only promoted by α-Ag1 for strain AB3879, but by α-Ag1, α-Ag5, and α-Ag7 for strain AB98 (**Fig 5G & 5H**). Overall, these data demonstrate that antibodies to the selected antigens bind to the surface of clinical *A. baumannii* strains, and this can result in C3b/iC3b opsonisation, and sometimes MAC formation, and can support neutrophil phagocytosis. However, the effects of a given antibody/ strain combination on complement activity or neutrophil phagocytosis often did not reflect the strength of the results for IgG opsonisation and whole cell ELISA data (summarised in **Table 3**), or the degree of immunofluorescence seen with microscopy for strain AB3879 (**Fig 3J**). The reasons for these discrepancies between immune assay results are unclear, but are likely related to differences in KL type, relative antigen expression between strains, and variations in the ability to promote complement activity between specific IgG to each target antigen.

## Rabbit antibodies to selected target antigens protected mice against *A. baumannii*

We tested the protective efficacy of IgG to our selected antigens initially using our established model of *A. baumannii* sepsis with the clinical MDR *A. baumannii* isolate AB3879 (KL10). This model requires co-infection with mucin to overcome the low virulence of many *A. baumannii* clinical isolates in immunocompetent mice [36,37]. To assess whether rabbit polyclonal antibody to Ag1, 5, 7, and 11 can promote protection against infection, CD1 mice were passively immunised by intraperitoneal inoculation of 50 μg/ mouse (approximately equivalent to 2 mg/ kg) with polyclonal rabbit IgG before

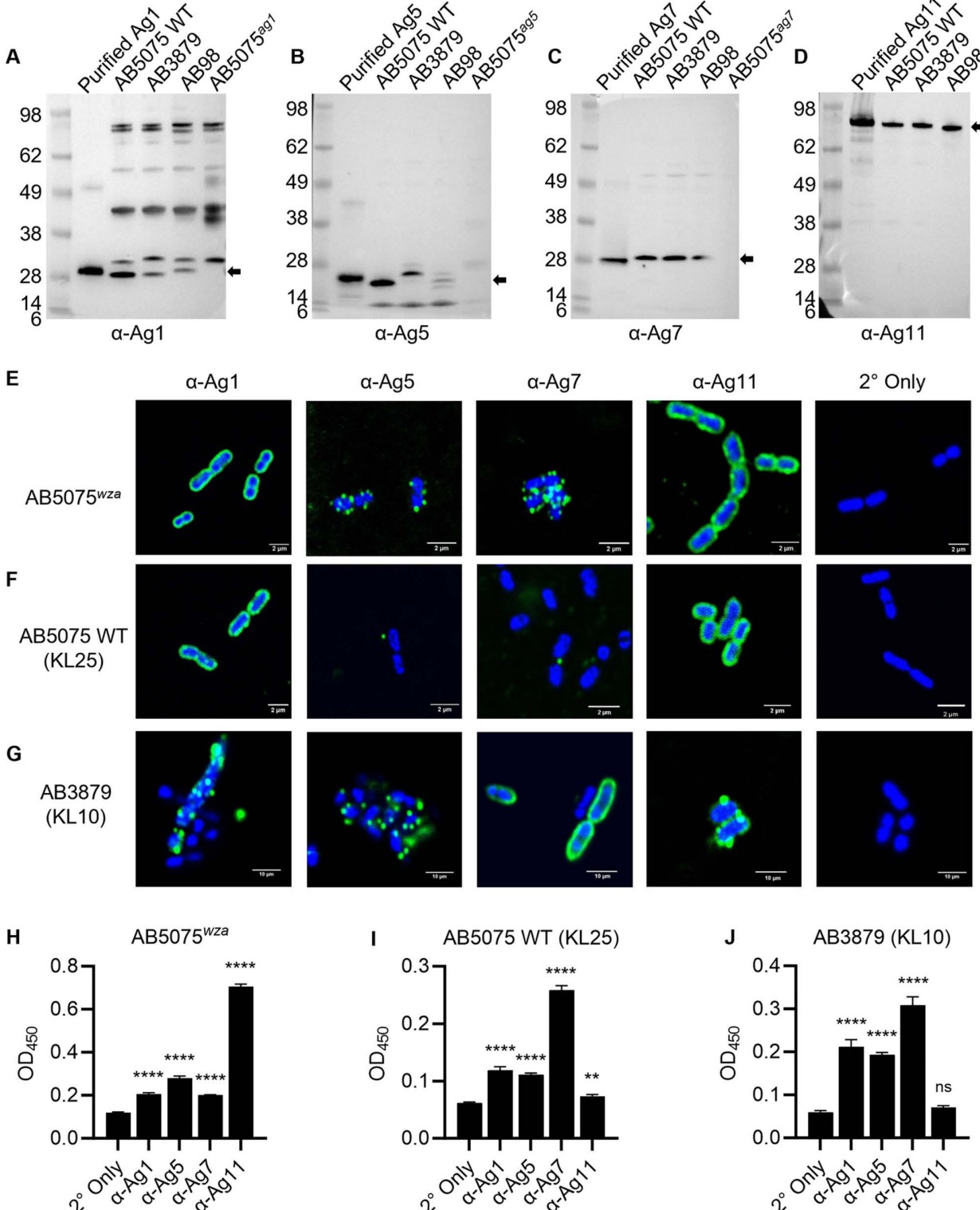

**Fig 3. Rabbit antibody recognition of selected antigens and clinical isolates. (A-D)** Immunoblots of purified protein antigens or lysates of indicated *A. baumannii* strains probed with rabbit polyclonal IgG to antigens 1, 5, 7, or 11. Molecular weight markers are labelled (kDa). Protein details including their predicted molecular weights (indicated by the black arrows) are shown in **Table 2**. **(E-G)** Fluorescent microscopy showing IgG deposition (green) of

indicated antibodies on *A. baumannii* for the unencapsulated mutant strain AB5075$^{wza}$ (E), the parental wild-type (WT) AB5075 strain (KL25) (F), and the encapsulated Thai clinical isolate AB3879 (KL10) (G). DNA is stained by DAPI (blue). (H-J) Mean ODs (error bars = SDs) for whole cell ELISAs for the AB5075$^{wza}$ (H), AB5075 WT (KL25) (I), and the AB3879 (KL10) (J) strains probed with 1:100 purified IgG antibody to the selected antigens, or the secondary antibody only control (2° only). KL; K locus capsule type. Data were analysed using ANOVA with Dunnett's multiple-comparison test (*$p < 0.05$, **$p < 0.01$, ***$p < 0.001$, ****$p < 0.0001$, ns; not significant) comparing to the 2° only control.

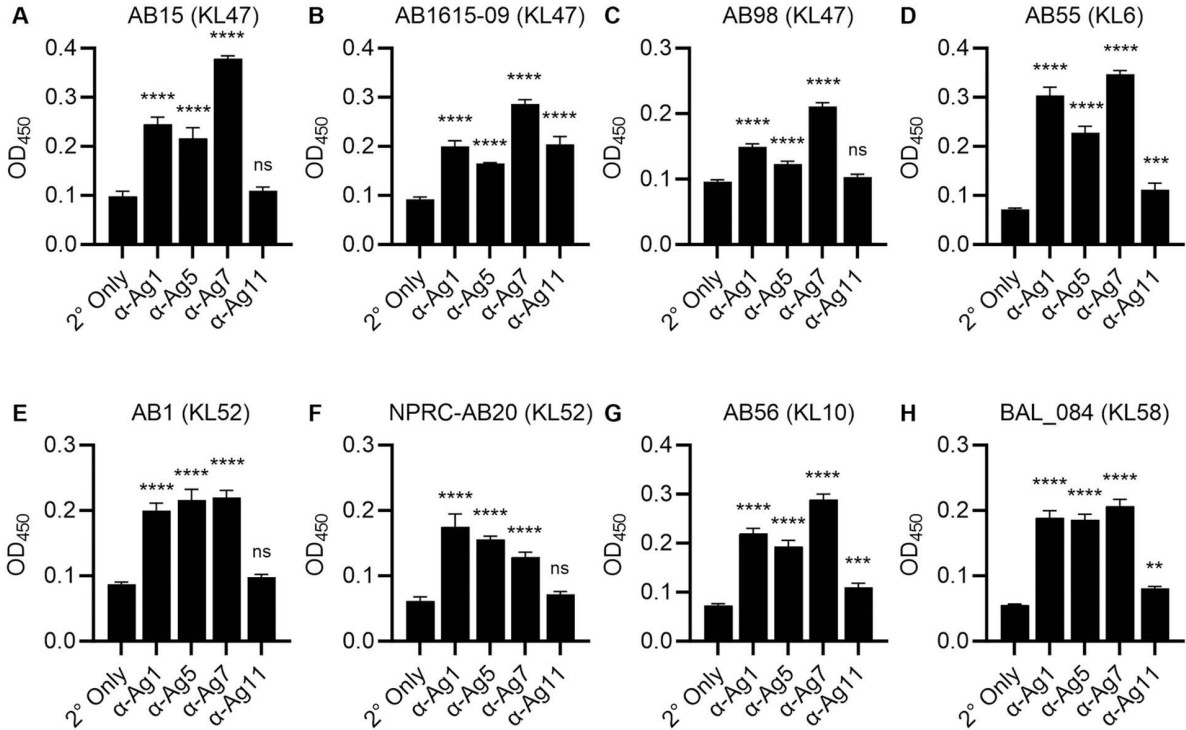

**Fig 4. Rabbit polyclonal IgG to the selected antigens recognises multiple clinical *A. baumannii* strains. (A-H)** Mean ODs (n = 4, error bars = SDs) for whole cell ELISAs for 8 clinical *A. baumannii* strains (named in each panel) [33] probed with 1:100 purified IgG polyclonal antibody to the selected antigens, or the secondary antibody only control (2° only). KL; K locus capsule type. Data were analysed using ANOVA with Dunnett's multiple-comparison test (*$p < 0.05$, **$p < 0.01$, ***$p < 0.001$, ****$p < 0.0001$, ns; not significant) comparing to the 2° only control.

intraperitoneal inoculation with 3–6 x 10⁶ CFU/mouse of AB3879 suspended in 5% mucin. Passive immunisation with α-Ag1, α-Ag5, α-Ag7, or α-Ag11 [25–27] significantly reduced bacterial CFU recovered from one or more target organs (spleen or kidney), with a high proportion of mice given α-Ag5, α-Ag7 and α-Ag11 having kidney or spleen CFU below the limit of detection (**Fig 6A**). To ensure protection was not related to antibody interaction with bacteria limited to within the intraperitoneal cavity, mice were passively immunised by intraperitoneal injection with 100 µg α-Ag1 or α-Ag11 prior to intravenous infection with strain AB3879. Both antibodies reduced spleen CFU and significantly enhanced clearance of *A. baumannii* in the blood, confirming protection by passive immunisation was dependent on antibody effects in the blood compartment (**Fig 6B**). Passive immunisation with polyclonal IgG antibodies to two intracellular proteins identified by the initial antigen screen (Ag9 = ABUW_0563 and Ag10 = ABUW_3426) failed to reduce target organ CFU in the AB3879 sepsis mode (**S1A and S1B Fig**), indicating that the protection provided by IgG to Ag1, 5, 7 and 11 was not due to non-specific effects of polyclonal rabbit IgG such as antibody to LPS. To assess whether protection could be achieved in a pneumonia model as well as for sepsis, passive immunisation with α-Ag1 and α-Ag11 experiments were repeated using a recently described model of *A. baumannii* lung infection using intranasal inoculation of BALB/c mice with 2 x 10⁷ CFU of

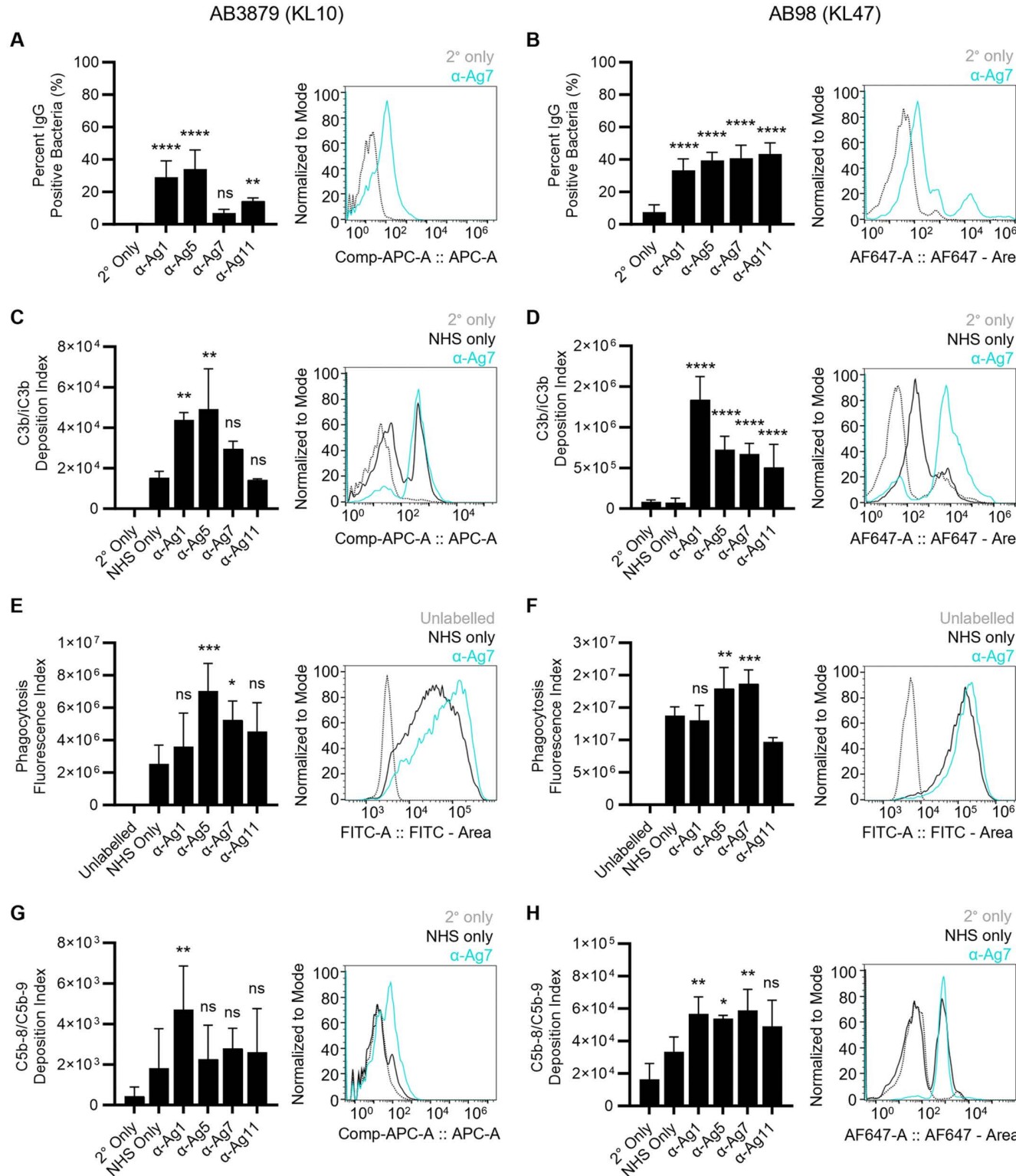

**Fig 5. *In vitro* flow cytometry assays of the effects of IgG to antigens 1, 5, 7, or 11 on opsonisation and complement recognition of *A. baumannii* strains AB3879 (KL10) (left panels) and AB98 (KL47) (right panels). (A)** and **(B)** Mean (n = 4, error bars = SDs) percent of bacteria positive for IgG after incubation with secondary antibody alone (2° only) or with antibody to *A. baumannii* Ag1, Ag5, Ag7, or Ag11. **(C)** and **(D)** Mean (SDs) C3b/iC3b

deposition index (% positive x median fluorescent intensity of bacteria staining positive for C3b/iC3b) after incubation in normal human serum (NHS only), or NHS plus antibody (as indicated), or the secondary antibody alone (2° only). **(E)** and **(F)** Bacterial association with fresh human neutrophils represented as the mean (SDs) phagocytosis index (% positive neutrophils x median fluorescent intensity of neutrophils staining positive for FAMSE-labelled *A. baumannii*) after incubation with NHS or NHS plus antibody. Unlabelled = non-FAMSE-labelled *A. baumannii* control. **(G)** and **(H)** Mean (SDs) C5b-8/C5b-9 deposition index (% positive x median fluorescent intensity of bacteria staining positive for C5b-8/C5b-9) after incubation in NHS, or NHS plus antibody, or the secondary antibody alone (2° only). For all panels, representative flow cytometry plots are shown after incubation with α-Ag7 (blue) compared to secondary antibody alone (grey dotted) **(A)** and **(B)**, or α-Ag7 plus NHS (blue) compared to NHS only controls (black solid) or secondary antibody alone (grey dotted) **(C-H)**. Data were analysed using ANOVA with Dunnett's multiple-comparison test (*$p < 0.05$, **$p < 0.01$, ***$p < 0.001$, ****$p < 0.0001$, ns; not significant) compared to secondary antibody controls **(A)** and **(B)** or compared to NHS only controls **(C-H)**. KL = K locus capsule type.

**Table 3. Summary of antibody efficacies in immune assays against two clinical *A. baumannii* isolates.**

| Antibody | Strain | ELISA$^{\alpha}$ | | IgG opsonisation$^{\beta}$ | | C3b/iC3b deposition$^{\delta}$ | | Neutrophil phagocytosis$^{\delta}$ | | MAC deposition$^{\delta}$ | |
|---|---|---|---|---|---|---|---|---|---|---|---|
| α-Ag1 | AB3879 | 0.21 | **** | 28.9% | **** | 2.8 | ** | 1.4 | ns | 2.6 | ** |
| | AB98 | 0.15 | **** | 33.2% | **** | 18.0 | **** | 1.1 | ns | 1.7 | ** |
| α-Ag5 | AB3879 | 0.19 | **** | 33.9% | **** | 3.2 | ** | 2.8 | *** | 1.2 | ns |
| | AB98 | 0.12 | **** | 39.5% | **** | 9.7 | **** | 1.3 | ** | 1.6 | * |
| α-Ag7 | AB3879 | 0.31 | **** | 6.8% | ns | 1.9 | ns | 2.1 | * | 1.5 | ns |
| | AB98 | 0.21 | **** | 40.9% | **** | 9.0 | **** | 1.4 | *** | 1.8 | ** |
| α-Ag11 | AB3879 | 0.07 | ns | 14.4% | ** | 0.9 | ns | 1.8 | ns | 1.4 | ns |
| | AB98 | 0.10 | ns | 43.4% | **** | 6.8 | **** | 0.7 | – | 1.5 | ns |

$^{\alpha}$ELISA OD$_{450}$ data (1:100 dilution).

$^{\beta}$Percent bacteria positive for IgG when assessed using flow cytometry.

$^{\delta}$ Fold change in complement deposition/phagocytosis compared to NHS only control.

*$p < 0.05$, **$p < 0.01$, ***$p < 0.001$, ****$p < 0.0001$, ns; not significant.

the Vietnamese AMR *A. baumannii* clinical strain BAL_084 (KL58) [38,39] 4 hours after intranasal inoculation of 100 μg/mouse of α-Ag1 or α-Ag11. By 24 hours after infection, lung CFU were reduced by approximately 1 log$_{10}$ in mice receiving either antibody compared to PBS controls (**Fig 6C**). To assess whether protection could be enhanced by administration of two antibodies in combination, mice were passively immunised with both α-Ag1 and α-Ag11 (the antibodies which showed the weakest level of protection in the single antibody experiments) before or after IP challenge with the AB3879 strain (**Fig 6D** and **6E**, respectively). Although passive immunisation with α-Ag1 in combination with α-Ag11 after infection did not reduce target organ *A. baumannii* CFU (**Fig 6E**), immunisation before infection strongly protected against infection and reduced target organ CFU to below the limit of detection for all mice, with evidence of additional benefit compared to immunisation with α-Ag1 or α-Ag11 alone (**Fig 6D**).

## Efficacy of selected antigens as active vaccine candidates

We then tested the antigens for their protective efficacy as active vaccination candidates. Two sets of mice were intraperitoneally immunised with Ag1, Ag5, Ag7, or Ag11 plus Sigma adjuvant. One set of mice were challenged with sepsis using strain AB3879 as above (**Fig 7A**), and the second set was used for sera collection to analyse antibody responses *in vitro* (**Fig 7B to 7D**). When vaccinated mice were challenged intraperitoneally with strain AB3879, mice vaccinated with Ag1 or Ag11 showed significant protection against infection (**Fig 7A**). Vaccinated mice generated an antigen-specific serum IgG response when measured using purified protein ELISAs (**Fig 7B**) but showed variable levels of IgG recognition of strain AB3879 when assessed using whole cell ELISAs (**Fig 7C**). Assessing IgG subtypes demonstrated that vaccination of mice with these antigens predominantly stimulated IgG1 and IgG2 subclasses (**Fig 7D**).

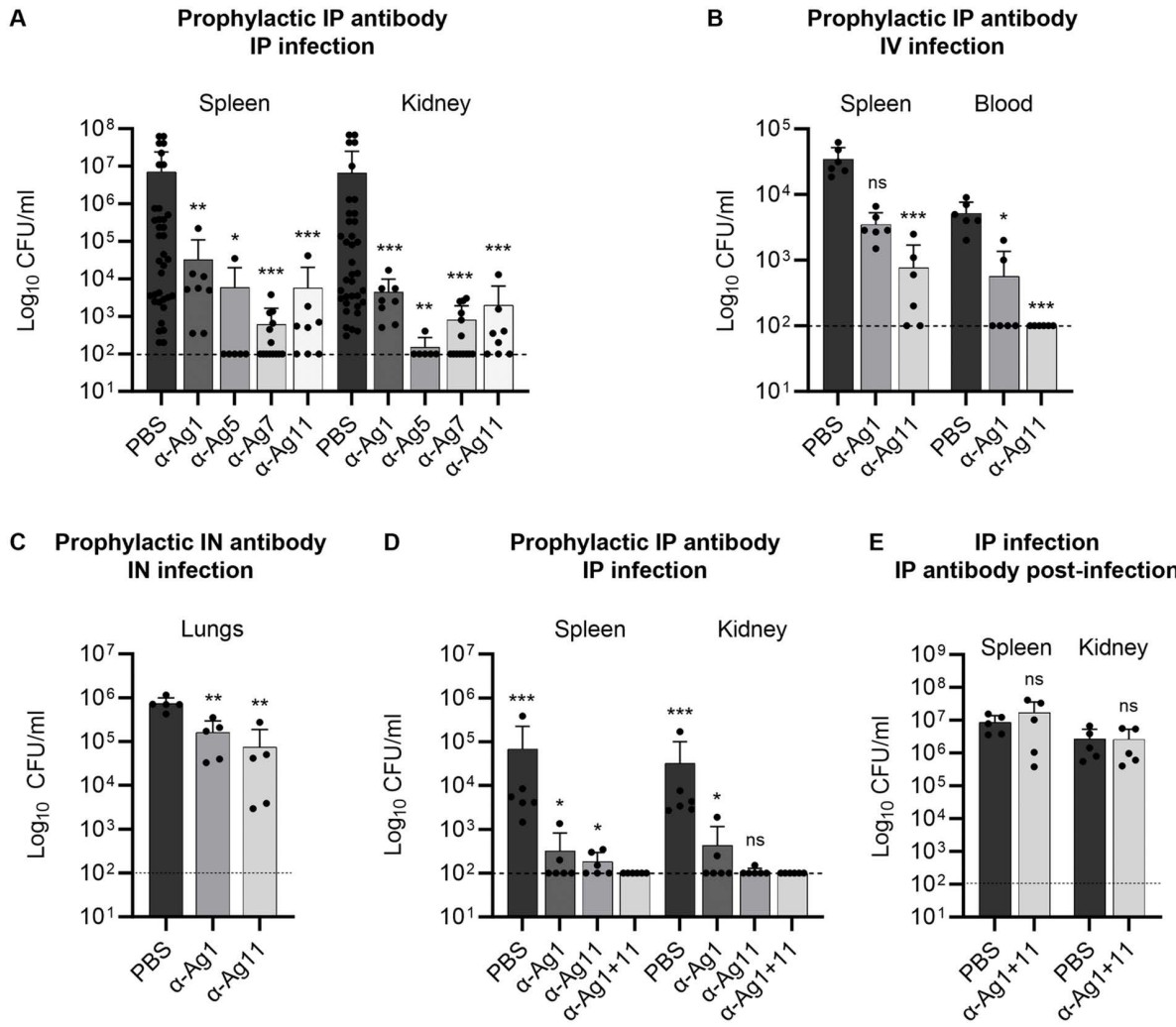

**Fig 6. Passive immunisation with rabbit IgG to the selected antigens in mouse models of *A. baumannii* infection. (A)** Mice (n ≥ 6) were passively immunised by intraperitoneal (IP) inoculation of 50 µg rabbit polyclonal IgG or PBS 4 h before IP inoculation with 3-6 x 10⁶ CFU/mouse of strain AB3879 (KL10) suspended in PBS with 5% porcine mucin. **(B)** Mice (n = 6) were passively immunised IP with 100 µg rabbit polyclonal IgG 4 h before intravenous (IV) infection with 2 x 10⁶ CFU/mouse of strain AB3879 (KL10) suspended in PBS. **(C)** Mice (n = 5) were passively immunised intranasally (IN) with 100 µg rabbit polyclonal IgG 4 h before IN infection with 2 x 10⁷ CFU/mouse of strain BAL_084 (KL58) suspended in PBS. **(D)** and **(E)** Mice (n = 5 **(D)**, n = 6 **(E)**) were passively immunised before **(D)** or after **(E)** IP infection with 6 x 10⁶ CFU/mouse of strain AB3879 (KL10) suspended in PBS with 5% porcine mucin using either PBS, 50 µg individual rabbit polyclonal IgG, or with a combination of 25 µg α-Ag1 + 25 µg α-Ag11, as indicated. Dot plots for all panels represent bacterial CFU in the indicated target organs from individual mice (bars = means, error bars = SDs) 20-24 h after IP and IN infection **(A, C, D, E, and F)**, or 2 h after IV infection **(B)**. Data were analysed using Kruskal-Wallis one-way analysis of variance compared to PBS controls except for data in **(D)** which was compared to the combination treatment (*$p < 0.05$, **$p < 0.01$, ***$p < 0.001$, ****$p < 0.0001$, ns; not significant). KL; K locus capsule type.

## Passive protection of mice by selected antibodies requires neutrophils and complement

To determine which phagocyte subset mediated passive protection with prophylactic rabbit antibody to our selected antigens in the AB3879 mouse sepsis model, infection experiments were repeated using α-Ag5 and α-Ag7 (the antibodies showing the strongest protection after passive immunisation in the sepsis model) after depletion of tissue resident macrophages or neutrophils. When using spleen CFU as the readout, treating mice with intraperitoneal clodronate that depleted splenic macrophages by 86.2% (SD 1.5%) (**Fig 8A**) did not prevent the protective effect of passive immunisation

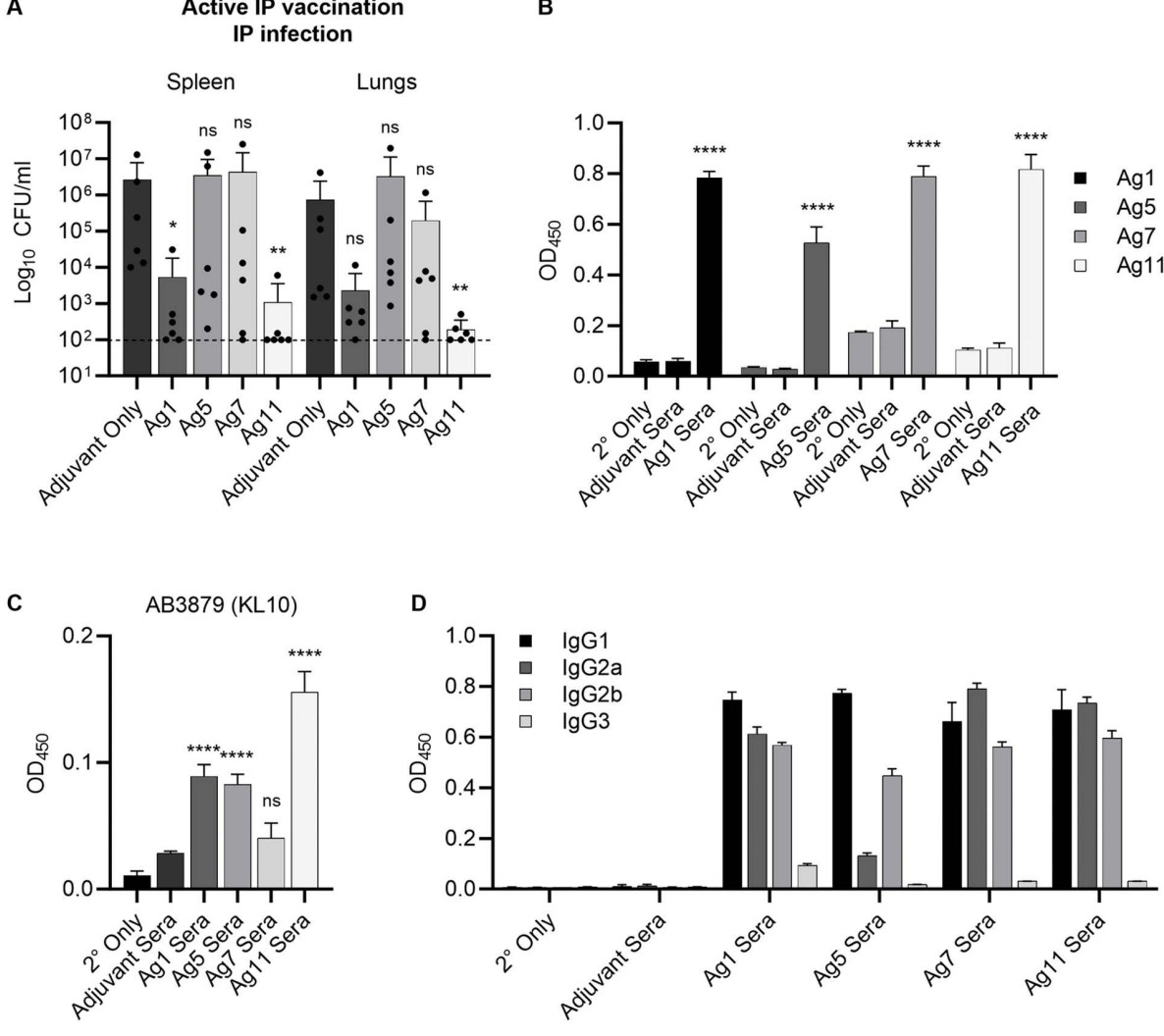

**Fig 7. Active vaccination of mice with selected protein antigens.** Two sets of mice (n = 6) were actively immunised intraperitoneally (IP) three times **(A)** or two times **(B-D)**, two weeks apart, with 5 µg purified protein or PBS, plus Sigma's adjuvant (1:1). **(A)** Four weeks following final vaccination, mice (n = 6) were infected IP with 3-6 x $10^6$ CFU/mouse of strain AB3879 (KL10) suspended in PBS with 5% porcine mucin. Dot plots represent bacterial CFU in the indicated target organs from individual mice (bars = means, error bars = SDs) 20-24 h after IP infection. Data were analysed using Kruskal-Wallis one-way analysis of variance compared to Adjuvant Only controls (*$p < 0.05$, **$p < 0.01$, ***$p < 0.001$, ****$p < 0.0001$, ns; not significant). **(B-D)** Sera were collected from uninfected mice (n = 6) two weeks after final vaccination. ELISA detection of purified protein antigens **(B)** or AB3879 whole cell lysates **(C)** using sera from mice vaccinated with respective antigens (Ag Sera) or Adjuvant only sera (Adjuvant Sera), or no sera control (2° Only). (Mouse sera were combined from 6 mice/group, bars = means of 4 replicates of combined sera for that group, error bars = SDs). **(D)** Purified protein ELISA to detect IgG subtypes in each vaccinated mouse sera. Indicated Ag sera were used to probe the respective purified protein Ags as in **(B)**, and indicated IgG subtype HRP-conjugated antibodies were used as a secondary antibody to detect IgG subtypes in vaccine sera. Ag1 purified protein was used for the adjuvant only sera (Adjuvant Sera) and no sera (2° only) controls. Statistical analysis compared to Adjuvant Only vaccine control sera used Dunnett's multiple comparison test (*$p < 0.05$, **$p < 0.01$, ***$p < 0.001$, ****$p < 0.0001$, ns; not significant).

with α-Ag5 or α-Ag7 (**Fig 8D**). In contrast, treating mice with intraperitoneal α-Ly6G to deplete neutrophils (reduced splenic neutrophils by 99.5%, SD 0.2%) (**Fig 8B**) prevented passive immunisation with α-Ag5 or α-Ag7 from protecting against sepsis (**Figs 8D** and **S2**), demonstrating that neutrophils were the main phagocyte mediating antibody-dependent protection in this model. To assess whether complement was also required, passive protection experiments were repeated after

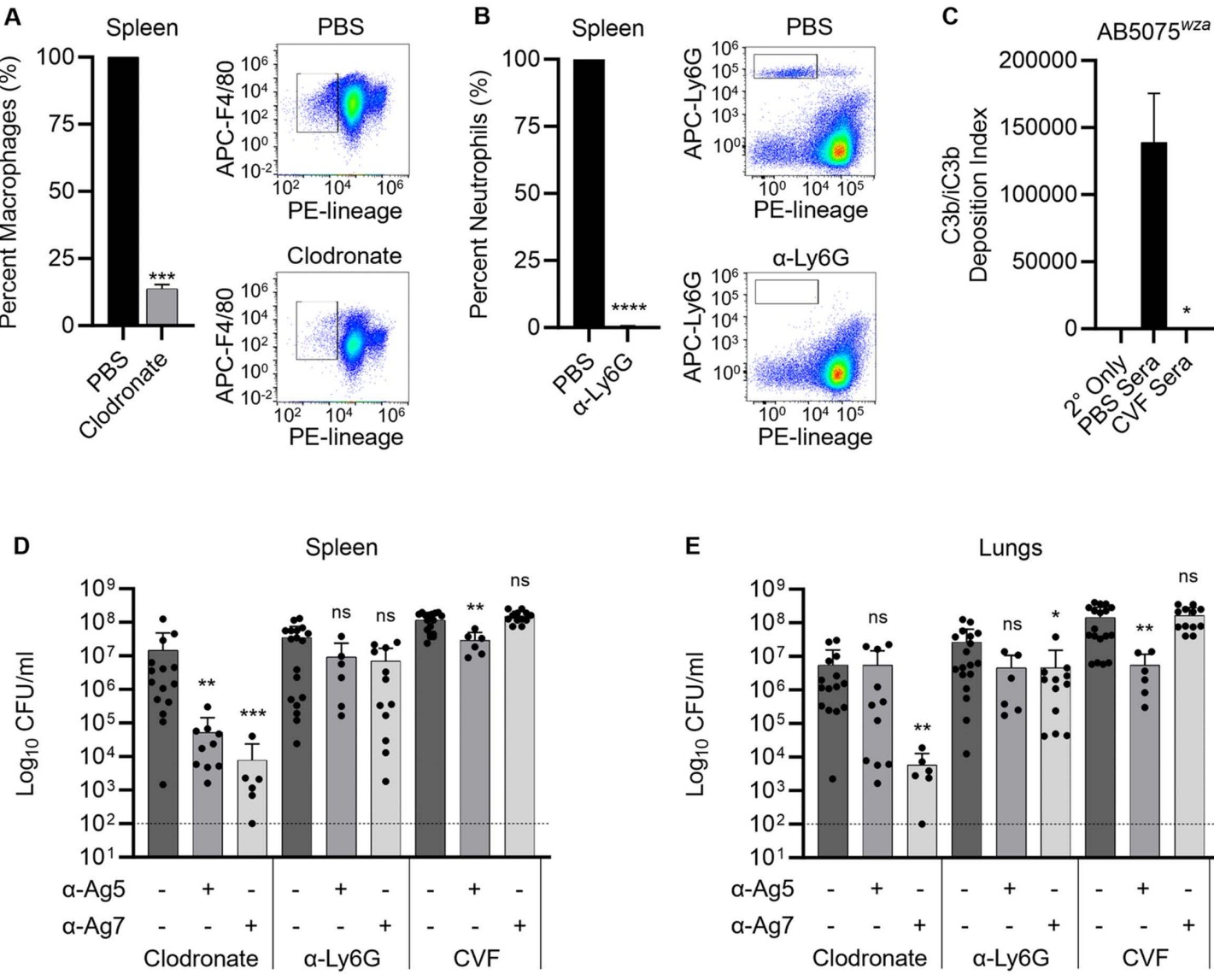

**Fig 8. Passive protection by rabbit antibodies to selected antigens requires C3b/iC3b opsonisation and neutrophils.** Mice were depleted of macrophages, neutrophils, or complement 24 or 48 h prior to passive immunization by intraperitoneal inoculation of 50 µg of rabbit polyclonal IgG or PBS. 4 h post-immunization mice were intraperitoneally inoculated with 3-6 x 10⁶ CFU/mouse of strain AB3879 (KL10) suspended in PBS with 5% porcine mucin. **(A-C)** Flow cytometry data showing percentages of mice splenic macrophages **(A)** or splenic neutrophils **(B)** after treatment with clodronate or α-Ly6G, respectively, compared to PBS treated mice. **(C)** Flow cytometry data showing the mean C3b/iC3b deposition index (% positive x median fluorescent intensity of bacteria staining positive for C3b/iC3b) on the unencapsulated *A. baumannii* strain AB5075$^{wza}$ after incubation with secondary antibody alone (2° only), or with sera from mice treated with PBS or with CVF to deplete complement (error bars = SDs). Statistical significance compared to PBS controls was determined using unpaired Student's *t* tests (*$p < 0.05$, **$p < 0.01$, ***$p < 0.001$, ****$p < 0.0001$, ns; not significant). **(D)** and **(E)** Dot plots represent bacterial CFU from individual mice (bars = means (n ≥ 6), error bars = SDs) in the spleens **(D)** or lungs **(E)** 20-24 h after infection. The data were analysed using Kruskal-Wallis one-way analysis of variance compared to CFUs recovered from depleted mice without antibody treatment (*$p < 0.05$, **$p < 0.01$, ***$p < 0.001$, ****$p < 0.0001$, ns; not significant). KL; K locus capsule type.

treating mice with cobra venom factor (CVF) which depletes serum complement. Complement depletion was confirmed by loss of C3b/iC3b deposition on unencapsulated *A. baumannii* strain AB5075$^{wza}$ incubated with CVF treated mouse serum (**Fig 8C**). When assessed using spleen CFU, CVF treatment abrogated the protective effect of passive immunisation with α-Ag5 and α-Ag7 against *A. baumannii* sepsis (**Fig 8D**), confirming complement is required for antibody mediated protection against *A. baumannii.* This is in line with data showing that polyclonal IgG antibodies to Ag9 and Ag10, which were unable to protect mice against sepsis infection, did not promote C3b deposition on clinical *A. baumannii* isolates (**S1 Fig**). The effects of depletion of macrophages, neutrophils, or C3b/iC3b on lung CFU in the sepsis model were more complex, with the protective effects of α-Ag7 dependent on neutrophils and complement but not macrophages (similar to the spleen data), whereas for α-Ag5 protection was dependent on both macrophages and neutrophils and partially on complement (**Fig 8E**).

## Discussion

AMR infections are a major global health threat, and novel therapies against MDR *A. baumannii* is a current global imperative [1,2,40–42]. Antibody-based therapies for AMR infections have significant potential advantages, including relatively fast speed of development and less potential for resistance by the pathogen [14]. Prophylactic antibody to prevent infection could be given to subjects at a high risk of infection, for example ventilated patients in Southeast Asia as these have a high incidence of *A. baumannii* pneumonia. Therapeutic antibody could also be given in combination with antibiotics to improve outcomes in patients with confirmed *A. baumannii* infection. The key to developing an effective antibody-based therapy is antigen target selection. The ideal antigen target would be surface exposed, highly conserved amongst *A. baumannii* strains causing infection, expressed during infection, and for which antibody recognition would improve immune recognition and promote bacterial clearance despite capsule-mediated impairment of access to subcapsular antigens [21,22]. Using a novel two-step methodology we have identified multiple protein antigen targets for *A. baumannii.* Candidate protein antigens were first identified by probing a protein microarray containing 868 conserved *A. baumannii* proteins with antisera obtained from mice after *A. baumannii* infections [35]. This was followed by *in vivo* and *in vitro* assays assessing the efficacy of prophylactic antibody to four selected candidate antigens at protecting against *A. baumannii* infection. Using this approach, we have shown that antibody to three novel protein antigens (a porin, a lipoprotein, and a peptidyl-prolyl isomerase) promoted immune recognition of multiple clinical encapsulated *A. baumannii* strains *in vitro* and protected against *A. baumannii* in mouse models of infection. Furthermore, active vaccine experiments showed vaccination with two of our antigens partially protected against *A. baumannii* sepsis. These data provide the basis for the development of vaccines or antibody therapies for use as prophylaxis against or as a treatment for MDR *A. baumannii* infections.

Our protein microarray contained 63% of the 1381 proteins that genome data suggest are highly conserved amongst 220 *A. baumannii* strains [20], including most of the proteins that *in silico* analysis predicted to be surface expressed. The microarray was enriched for proteins that are likely to be expressed during systemic infection, as indicated by *ex vivo* serum expression data. Since the total number of predicted surface proteins is fewer than the spaces available on the microarray, and since the prediction of surface localisation for bacterial proteins can be incorrect, some conserved cytoplasmic proteins were also included in the array. Inclusion of these antigens will also broaden the data obtained for planned future studies using the microarray to measure human serological responses to *A. baumannii* infection. Antisera for probing the microarray was raised by non-lethal infection in mice, which we have previously shown induces IgG to multiple protein antigens that can mediate serotype-independent protection [33]. An interesting observation is that using three different *A. baumannii* strains to raise antisera resulted in IgG recognition of many more proteins compared to three inoculations of the same strain, perhaps due to restricted expression of many proteins by any single strain. In total, 66 proteins (7.6% of proteins on the microarray) were recognised by IgG in mouse sera. Further work could identify additional antigenic proteins by probing the microarray with sera from humans recovering from *A. baumannii* infection or with sera obtained from mice infected with other *A. baumannii* strains.

Three previously undescribed *A. baumannii* proteins and the known protective antigen BamA [25–27] were selected from the microarray data for further investigation. As yet there are only minimal data on the potential functions of Ag1, 5 and 7. Ag1 is a predicted porin that may play a role in the transport of nutrients into the cell or efflux of antibiotics and hence may contribute to virulence [43–46]. Ag5 is a predicted lipoprotein that is currently uncharacterised. Bacterial lipoproteins frequently contribute to host-pathogen interactions and antibiotic resistance [47], and in *A. baumannii* can affect cellular morphology, serum resistance and resistance to phagocytosis [48]. Additionally, licensed protein-based vaccines against *Neisseria meningitidis* include Factor H-binding lipoproteins [49,50], further demonstrating the potential of these lipoproteins as antigenic therapeutic targets. Finally, Ag7 is a predicted FK05-binding protein-type peptidyl-prolyl *cis-trans* isomerase (FkpA), the gene for which is located next to the *A. baumannii* capsule gene cluster [51]. This protein has 46% identity to the *Pseudomonas aeruginosa* chaperone protein PaFkbA that assists protein folding [52]. Molecular chaperones involved in protein folding, assembly, and trafficking are essential to bacterial stress responses, especially when adapting to the host environment [53]. Overall, targeting our selected antigens with antibodies could not only support immune mediated clearance but potentially may also impact bacterial cellular functions affecting virulence and sensitivity to antibiotics.

*In vitro* assays demonstrated that rabbit antibody to the four selected novel protein antigens recognised multiple clinical AMR *A. baumannii* strains. Antibody to all four selected protein antigens partially protected against *A. baumannii* infection in mice when given prior to infection, with significant reductions in target organ CFU varying from around 3 to approaching 5 $\log_{10}$ CFU, depending on antibody and target organ. To improve clinical relevance, our sepsis model uses an MDR clinical isolate of *A. baumannii* which requires co-administration with mucin to ensure infection is established [36,37] but can result in significant variability in target organ CFU in PBS treated mice. Despite this variability reducing statistical power, administration of polyclonal IgG to our novel target antigens reproducibly reduced target organ CFU to a similar or lower level than the BamA positive control. Smaller but significant reductions in lung CFU were seen in a separate pneumonia model using a Vietnamese clinical *A. baumannii* strain which does not require coadministration of mucin. Overall, the protective efficacy of polyclonal IgG to or active vaccination with our target antigens against *A. baumannii* infection was similar to that seen with vaccines or antibody therapies targeting BamA [25,27] and other protein antigens such as OmpA [54,55], Oxa-23 [16], the IROMPs BauA and BfnA [31,55], Ata [56], and SmpA [57].

Antibody to capsular antigen has stronger protective efficacy compared to antibody to protein antigens due to the high quantity of surface accessible target antigen [16,17,21–23,33] and capsular antibodies can protect mice against *A. baumannii* when administered after infection was initiated [16]. In contrast, the capsule partially shields protein antigens from antibody and this impairs protective efficacy [16,17,21–23,33]. Passive immunisation with polyclonal antibody to our target antigens was only protective if administered before inoculation of *A. baumannii*. Furthermore, in common with many mouse models of *A. baumannii* infection [25,26,28,32,54–61], our infection models progress rapidly often resulting in very high levels of *A. baumannii* CFU in target organs within 24 hours, and this also would make post-infection administration of antibody less effective. In contrast, *A. baumannii* infections in humans are generally slowly progressive and patients will also be receiving antibiotics [62], both of which would be likely to make treatment with an antibody therapy more efficacious. Furthermore, antibody engineering and targeting two or more antigens could improve the protective efficacy of antibody to protein antigens so they can be used as a potential treatment for proven *A. baumannii* infections: indeed, our data show that passive vaccination with antibody to two antigens was more efficacious than targeting a single antigen, supporting a multivalent approach and the need to identify multiple suitable protein antigen targets.

Antibodies to the selected antigens promoted immune recognition of different *A. baumannii* strains when assessed using *in vitro* assays of bacterial opsonisation with IgG or C3b/iC3b, MAC formation, and neutrophil phagocytosis. The degree of improved opsonisation with IgG or C3b/iC3b varied between antibodies and between strains, potentially representing the effects of differential antigen expression, antibody accessibility to the protein target, and variations between *A. baumannii* strains in their intrinsic resistance to complement or phagocytosis [33,35]. Levels

of opsonisation with IgG or C3b/iC3b, MAC deposition, and neutrophil phagocytosis might be expected to correlate closely, but this was not always the case. Potentially this reflects the complex interplay between relative antigen expression by each strain, the effects of the capsule and other bacterial factors on differences in strain/ antibody immune interactions, along with variations between polyclonal antibodies in their avidity to target antigens and ability to activate complement or promote recognition by neutrophil receptors [21,33,35]. Characterising these effects would be complex and require otherwise isogenic strains engineered to express different KL types. Overall, although the *in vitro* assays confirm a given antibody will promote immune recognition of *A. baumannii*, the lack of clear correlations in the results for different immune assays makes identifying the most effective candidate target antigen using *in vitro* immune assays difficult.

Antibody to *A. baumannii* could promote protection through increasing opsonisation of the bacteria with IgG and/ or complement, thereby improving neutrophil and/ or macrophage mediated opsonophagocytosis, or through promoting bacterial agglutination that also improves phagocytosis and impairs bacterial growth. Furthermore, antibody-mediated increased complement recognition of *A. baumannii* could lead to direct killing by MAC formation. Passive vaccination with polyclonal antibody to Ag1 or Ag11 promoted rapid reductions in blood CFU after intravenous inoculation, compatible with the antibody increasing the speed of bacterial clearance by the innate immune response. This was confirmed by additional experiments in mice depleted of phagocytes or complement, which showed that the protective efficacy of antibodies to Ag5 or Ag7 was mediated by complement and neutrophils but not macrophages. This is in line with other studies demonstrating that neutrophils play a key role in innate [63] and mAb-mediated [64] protection. Overall, our results indicate that the protective antibodies promoted bacterial opsonisation with C3b/iC3b during mouse infection, thereby partially overcoming capsule-dependent inhibition of complement recognition of *A. baumannii* [20,35,65], and improving neutrophil-dependent killing. Hence, antibody engineering to improve complement activation should improve the protective efficacy of antibody to our protein antigens. Systemic macrophage-mediated clearance of *A. baumannii* in mice appeared to be largely independent of complement activity and instead might be dependent on direct binding of cell surface receptors to bacterial ligands, such as the capsule, as has been described for *Streptococcus pneumoniae* and *Klebsiella pneumoniae* [66,67]. How co-inoculation of *A. baumannii* with mucin enhances virulence is not fully understood and theoretically could include altering the relative contributions of macrophages to neutrophils or complement to immune clearance, thereby skewing which immune effector mediates protection of antibody to our antigens. However, depletion of peritoneal macrophages does not abrogate mucin-dependent enhancement of virulence, which instead seems to be partially dependent on increasing iron availability [36], suggesting our data on the role of neutrophil and complement would be relevant for *A. baumannii* infection models that are not dependent on coadministration of mucin.

To conclude, using a novel protein microarray screening methodology we have identified 66 antigenic *A. baumannii* proteins, and demonstrated that antibody to four of these promoted immune recognition of different clinical *A. baumannii* strains and protection in mouse models. These data will aid the development of immunotherapeutics, including vaccines or mAbs to prevent or treat infection, that can reduce the substantial and increasing morbidity and mortality caused by AMR *A. baumannii* infections. Using a similar experimental approach could also identify conserved protein targets for antibody therapies or vaccines for multiple other AMR bacterial pathogens.

## Materials and methods

### Ethics statement

All mouse experiments were performed at University College London following national guidelines and were approved by the University College London Biological Services Ethical Committee and the UK Home Office (project number PP3031297).

 

## Bacterial strains and culture conditions

All bacterial strains used for this study are listed in **S2 Table**. Clinical *A. baumannii* isolates were isolated from patients admitted to hospital in Thailand [33] or Vietnam [38,39]. These strains are available to the wider research community upon request. Strains AB5075, AB5075*wza*, AB5075*ag5*, and AB5075*ag7* were obtained from the Manoil laboratory *A. baumannii* mutant library [68]. Strain AB5075*ag1* is a deletion mutant and was generated in this study using suicide plasmid-mediated mutagenesis as described in [69]. Briefly, an approximately 2 kb DNA fragment containing mutated *ABUW_0166* was synthesized by GenScript. The PCR fragment was inserted into the suicide plasmid pSL15A (Manoil laboratory) via restriction digestion and ligation, and the resulting plasmid (pSL15A_*Δ0166*) was cloned into *Escherichia coli* strain MFDpir [70] using heat shock transformation. The suicide plasmid was then transferred from *E. coli* to *A. baumannii* strain AB5075 by conjugation. The final *A. baumannii* AB5075*ag1* mutant was generated by selection on sucrose as described previously [69] and confirmed using PCR amplification and DNA sequencing of the mutated chromosomal DNA (Source BioScience). Bacteria were plated and cultured in Luria-Bertani broth (LB, Lennox formulation, Sigma). Stocks of each strain containing 15% glycerol (v/v) were stored at -70°C. Unless otherwise specified, cultures were incubated at 37°C with shaking at 200 rpm. Bacterial growth at $OD_{600}$ was measured over 24 h using the Tecan Spark multimode microplate reader [33,35].

## RNAseq experiments

Transcriptome analyses by RNAseq were previously described [35,71]. Briefly, strains AB1615-09, AB1, and NPRC-AB20 were grown in triplicate in 50% normal human serum (NHS) or 50% LB for 1 h, then RNA extracted using the Mirvana RNA kit (Applied Biosystems) as previously described [35]. RNA sequencing libraries were processed using the KAPA RNA HyperPrep kit, the data mapped to the AB1615-09 genome and transcripts quantified as described [35]. Raw RNAseq data were uploaded to the European Nucleotide Archive (ENA) and the individual data file accession numbers are as follows: ERR8982504, ERR8982505, ERR8982506, ERR8982507, ERR8982508, ERR8982509, ERR8982510, ERR8982511, ERR8982512, ERR8982513, ERR8982514, ERR8982515, ERR8982516, ERR8982517, ERR8982518, ERR8982519, ERR8982520, ERR8982521 [35].

## Design and probing of protein microarray

The core genome of 220 *A. baumannii* strains [20] was determined by Roary [72], and cellular locations of the corresponding proteins modelled with PSORT (v3.0) using strain AB1615-09 derived protein sequences. Differential RNAseq expression data from three Thai *A. baumannii* strains exposed to serum [35] were combined with the core gene and cellular localisation data to select 868 proteins for construction of protein microarrays using an *Escherichia coli in vitro* transcription and translation (IVTT) expression system [73,74]. Protein microarray pads were rehydrated in Fast Protein Array Blocking Buffer for 30 min and probed as previously described [74,75]. Briefly, sera samples were diluted in 10x His peptide (10 μg/μl sera) and 10% *E. coli* lysate blocking buffer, incubated for 30 min at room temperature, then added to the protein microarrays and incubated with gentle agitation at 4°C overnight. After washing in 1 x TBST (0.1% Tween 20) at room temperature, biotin-conjugated goat α-mouse IgG (BioLegend) in blocking buffer was added for 1 h at room temperature, followed by washing in 1 x TBST and addition of Streptavidin Alexa Fluor 647 conjugate (Invitrogen) for 1 h at room temperature, rinsed with water, dried and visualised using an Innopsys InnoScan 710 Microarray Scanner.

## Production of selected antigens and antibodies, and *In Vitro* immune assays

Recombinant proteins were produced in *E. coli* strain BL21 Star (DE3) by GenScript. Polyclonal rabbit IgG to selected protein antigens were obtained from ThermoFisher Scientific using their standard 90-day rabbit immunisation protocol; briefly, rabbits were immunised subcutaneously using Complete Freund's Adjuvant or Incomplete Freund's Adjuvant on day 1, 14, 42, and 56, with bleeds collected on days 0, 28, 56, and 72, and IgG purified using Protein A/G purification.

For each GenScript protein preparation, as expected LPS contamination was detected (proteins were diluted 1:4 and LPS was measured using the Pierce Chromogenic Endotoxin Quant Kit; Ag1 = 3.15 ± 0.16 EU/ml, Ag5 = 3.10 ± 0.21 EU/ml, Ag7 = 3.22 ± 0.15 EU/ml, Ag11 = 3.18 ± 0.05 EU/ml). Western immunoblots using denaturing gels on *A. baumannii* whole cell lysates or purified proteins were performed as described previously [69] using Novex sample reducing agent and Novex sample buffer, 4–12% Tris-Glycine Novex WedgeWell polyacrylamide gels (ThermoFisher Scientific) transferred to nitrocellulose membranes using the iBlot 2 Dry Blotting System (ThermoFisher Scientific), then probed with horseradish peroxidase-conjugated goat α-rabbit or α-mouse IgG antibody (Abcam) and visualised using chemiluminescent ECL reagent (Amersham) and an ImageQuant LAS 4000. Whole Cell ELISAs and confocal microscopy of *A. baumannii* were performed as described [33], using Alexa Fluor Plus 488 goat α-rabbit IgG (H + L) (Invitrogen) and Prolong Gold Antifade Mountant with DAPI (Invitrogen) for confocal microscopy [35]. Slides were imaged using a Zeiss LSM 800 confocal microscope and analysed using ImageJ 1.53t software. Flow Cytometry assays of IgG opsonisation (Alexa Fluor 647 donkey α-rabbit IgG, BioLegend), C3b/iC3b deposition (mAb clone 6C9 BioLegend), C5b-8/C5b-9 formation (mAb clone aE11 Abcam) on *A. baumannii* or neutrophil association (using fresh human neutrophils and fluorescent *A. baumannii*) were performed as previously described [33,35] and analysed using a BD FACSVerse or Sony ID7000 Spectral Analyser and FlowJo software for Windows (Version 10).

### Animal studies

Mice were obtained from Charles River Laboratories UK. α-*A. baumannii* mouse sera for probing microarrays were generated by intraperitoneal inoculation of 6–8-week-old CD1 mice with 1 x 10$^6$ CFU three times 14 days apart [33] with mice culled 4 weeks after the last immunisation.

For passive immunisation and challenge experiments, 6–8-week-old CD1 mice (females and males) were immunised intraperitoneally (sepsis models) or intranasally (pneumonia model) with 100 µl 0.5-1.0 mg/ml or 50 µl 2 mg/ml, (respectively) purified IgG rabbit antibody either 4 h before or 2 h after infection (as indicated in figure legends). Mice were infected by intraperitoneal inoculation of ~3–6 x 10$^6$ CFU/mouse of strain AB3879 suspended in PBS with 5% porcine mucin (Sigma), or by intravenous infection with 2 x 10$^6$ CFU/mouse of strain AB3879 in PBS, or by intranasal inoculation of 2 x 10$^7$ CFU/mouse of strain BAL_084 in PBS, under general anaesthesia. Bacterial CFUs in spleens, kidneys, blood, and lungs were determined either 2 h (IV clearance model) or 20–24 h (sepsis and pneumonia models) post-infection by plating serial dilutions of organ homogenates onto LB agar.

For active immunisation and challenge experiments, two groups of 6–8-week-old CD1 female mice were immunised intraperitoneally two (sera group) or three (challenge group) times, two weeks apart, with 5 µg purified protein, or PBS, plus Sigma's adjuvant (1:1). Two weeks following final vaccination, sera were collected form one group of mice for *in vitro* analysis. Four weeks following final vaccination of the second group, mice were infected intraperitoneally (sepsis model) with 2 x 10$^7$ CFU/mouse of strain AB3879 suspended in PBS with 5% porcine mucin (Sigma). Bacterial CFUs in spleens and lungs were determined 20–24 h post-infection by plating serial dilutions of organ homogenates onto LB agar.

For immune depletion experiments, female mice were injected intraperitoneally with 100 µl Clophosome Clodronate Liposomes (Neutral) (Stratech) 48 h pre-infection to deplete tissue macrophages, 500 µg α-mouse Ly6G (clone 1A8, 2B Scientific) 24 h pre-infection to deplete neutrophils (or rat IgG2a isotype control, anti-trinitrophenol (clone 2A3, 2B Scientific)), or 30 µg Cobra Venom Factor (CVF; Quidel) 24 h pre-infection to deplete complement. Immune cell depletion was confirmed by flow cytometry of homogenised spleens using Zombie UV Fixable Viability Kit (BioLegend) and the following antibodies (BioLegend); FITC α-mouse CD45 (S18009F), PE α-mouse CD3 (17A2), PE α-mouse CD19 (1D3/CD19), PE α-mouse NK1.1 (PK136), PE α-mouse CD170 (Siglec-F) (S17007L), PE α-mouse CD11c (N418), PE α-mouse Ly6G (Clone 1A8) or PerCP α-mouse Ly6G (HK1.4), Brilliant Violet 510 α-mouse CD11b (M1/70), and PE/Cy7 α-mouse F4/80 (BM8). Viable macrophages were identified as CD45$^+$ CD3$^-$ CD19$^-$ NK1.1$^-$ CD170$^-$ Ly6G$^-$ F4/80$^{high}$. Viable neutrophils were

identified as CD45$^+$ CD3$^-$ CD19$^-$ NK1.1$^-$ CD170$^-$ CD11c$^-$ F4/80$^-$ CD11b$^+$ Ly6G$^{high}$. Samples were run on a Sony ID7000 Spectral Analyser and analysed using FlowJo software for Windows (Version 10).

## Statistical analysis

Microarray analysis was performed by generating a Threshold MFI value for each antigen (Threshold = (average "no sera" MFI + (2*SD of "no sera" MFI)) + (average "naïve sera" MFI + (2*SD of "naïve sera" MFI)) which was then subtracted from the raw MFI value. These corrected values were then averaged across sera to produce average MFI values +/- SDs (**S1 Table**). Antigen recognition was classed as significant if the mean corrected MFI was greater than the antigen threshold value and >500. Statistical analyses for all experiments were performed using GraphPad Prism Version 9.3.1 (GraphPad, USA). *In vitro* data represent at least two independent experimental repeats, are presented as means of at least three replicates, and error bars represent standard deviations. Mouse infection experiments showing significant differences between antibody and PBS groups were repeated at least twice to ensure the data were reproducible. Individual statistical tests are indicated in figure legends.

## Supporting information

**S1 Fig. Effects of less protective rabbit IgG to selected antigens. (A and B)** Mice (n = 6) were passively immunised by intraperitoneal (IP) inoculation of 50 μg rabbit polyclonal IgG or PBS 4 h before IP inoculation with 3–6 x 10$^6$ CFU/mouse of strain AB3879 suspended in PBS with 5% porcine mucin. Dot plots represent bacterial CFU in the indicated target organs from individual mice (bars = means, error bars = SDs) 20–24 h after IP infection. Data were analysed using Kruskal-Wallis one-way analysis of variance compared to PBS controls (*$p < 0.05$, **$p < 0.01$, ***$p < 0.001$, ****$p < 0.0001$, ns; not significant). **(C and D)** Mean (SDs) C3b/iC3b deposition index (% positive x median fluorescent intensity of bacteria staining positive for C3b/iC3b) after incubation in normal human serum (NHS only), or NHS plus antibody (as indicated), or the secondary antibody alone (2° Only).
(DOCX)

**S2 Fig. Neutrophil depletion prevents passive protection by rabbit antibodies to selected antigens.** Mice (n = 6) were injected intraperitoneally (IP) with PBS, α-Ly6G, or an isotype control antibody 24 h prior to passive immunization by IP inoculation of 50 μg of rabbit polyclonal IgG to Ag7, or PBS. 4 h post-immunization mice were inoculated IP with 3–6 x 10$^6$ CFU/mouse of strain AB3879 suspended in PBS with 5% porcine mucin. Dot plots represent bacterial CFU in the indicated target organs from individual mice (bars = means, error bars = SDs) 20–24 h after IP infection. Data were analysed using Kruskal-Wallis one-way analysis of variance compared to no treatment controls (*$p < 0.05$, **$p < 0.01$, ***$p < 0.001$, ****$p < 0.0001$, ns; not significant).
(DOCX)

**S3 Fig. Colistin resistance data for clinical *A. baumannii* isolates.** Growth of *A. baumannii* clinical isolates over 24 h at 37°C in LB media **(A)** or LB plus 4 μg/ml colistin **(B)** (means, error bars = SDs, n = 3). MIC ≥ 4 μg/ml = resistant to colistin in accordance with CLSI MIC Breakpoints 2023. KL = K locus capsule type.
(DOCX)

**S1 Table. Genome conservation data on selected microarray proteins, their surface localisation, RNAseq expression in human serum, and microarray data.**
(XLSX)

**S2 Table. *Acinetobacter baumannii* clinical isolates and their antibiotic sensitivities (R = resistant).**
(DOCX)

**S3 Table. *Acinetobacter baumannii* strains used for genome analysis of gene conservation.**
(XLSX)

## Author contributions

**Conceptualization:** Samantha Palethorpe, Giuseppe Ercoli, Elisa Ramos-Sevillano, Gathoni Kamuyu, Brendan Wren, Ganjana Lertmemongkolchai, Richard Stabler, Jeremy S. Brown.

**Data curation:** Samantha Palethorpe, Giuseppe Ercoli, Elisa Ramos-Sevillano, Gathoni Kamuyu, Samuel Willcocks.

**Formal analysis:** Samantha Palethorpe, Joe Campo, Samuel Willcocks, Philip Felgner, Richard Stabler.

**Funding acquisition:** Brendan Wren, Ganjana Lertmemongkolchai, Jeremy S. Brown.

**Investigation:** Samantha Palethorpe, Giuseppe Ercoli, Elisa Ramos-Sevillano, Gathoni Kamuyu, Samuel Willcocks, Rie Nakajima, Philip Felgner, Richard Stabler.

**Methodology:** Samantha Palethorpe, Giuseppe Ercoli, Elisa Ramos-Sevillano, Gathoni Kamuyu, Samuel Willcocks, Rie Nakajima, Philip Felgner.

**Supervision:** Jeremy S. Brown.

**Writing – original draft:** Samantha Palethorpe, Jeremy S. Brown.

**Writing – review & editing:** Samantha Palethorpe, Brendan Wren, Ganjana Lertmemongkolchai, Richard Stabler, Jeremy S. Brown.

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
