## [Decision Letter · Decision Letter 0]

7 Oct 2025

Identification of multiple *Acinetobacter baumannii* protein antigens as targets for potential immunotherapies using a novel protein microarray screening approach

PLOS Pathogens

Dear Dr. Brown,

Thank you for submitting your manuscript to PLOS Pathogens. Again, we sincerely apologize for the delay in the review process. Your manuscript was evaluated by members of the editorial board, a guest editor, and three external referees. All believe that your study is interesting and important, but have concerns that need to be addressed. Therefore, we invite you to submit a substantially revised version of the manuscript that addresses all of the points raised by the three reviewers. In particular, you need additional discussion of the biological functions of the antigens, clarification of experimental details, controls that include whether LPS is present in the vaccine preparations, and publicly available RNA-seq data..

Please submit your revised manuscript within 60 days Dec 06 2025 11:59PM. If you will need more time than this to complete your revisions, please reply to this message or contact the journal office at plospathogens@plos.org. Please include the following items when submitting your revised manuscript:

We look forward to receiving your revised manuscript.

Kind regards,

Lauren D. Palmer, PhD

Guest Editor

PLOS Pathogens

D. Scott Samuels

Section Editor

PLOS Pathogens

Sumita Bhaduri-McIntosh

Editor-in-Chief

PLOS Pathogens

orcid.org/0000-0003-2946-9497

Michael Malim

Editor-in-Chief

PLOS Pathogens

**Journal Requirements:**

1) Please provide an Author Summary. This should appear in your manuscript between the Abstract (if applicable) and the Introduction, and should be 150-200 words long. The aim should be to make your findings accessible to a wide audience that includes both scientists and non-scientists. Sample summaries can be found on our website under Submission Guidelines:

https://journals.plos.org/plospathogens/s/submission-guidelines#loc-parts-of-a-submission

3) Please amend your detailed Financial Disclosure statement. This is published with the article. It must therefore be completed in full sentences and contain the exact wording you wish to be published.

**Reviewers' Comments:**

Reviewer's Responses to Questions

**Part I - Summary**

Reviewer #1: Overall this is a very interesting study that identifies multiple novel antigens and evaluates their potential as vaccine candidates. A key part of the novelty lies in the methodology where a novel protein microarray approach was used. Three candidates from this array were taken forward for further characterisation alongside a positive control in BamA which was been used extensively in candidate vaccine studies. Polyclonal antibodies are raised to these novel vaccine candidates and used to demonstrate cellular localisation of the targets but also that some can be targeted in a capsulated cell (Ag-1 and Ag-11) while others (Ag-5/7) have limited binding in capsulated cells. However, they both performed better in a capsulated Thai clinical isolate (Ab3879) which suggests KL type will influence efficacy, as expected. ELISAs were also used to confirm recognition across strains with α-Ag-1 showing the highest titre. The robust mouse studies which include multiple infection models are a particular highlight of the study as they offer robust in vivo validation of the potential efficacy of these candidates. The study overall is well structured with a clear rationale throughout and a strong supporting narrative.

Reviewer #2: The manuscript is well-aligned with its title, and the data presented strongly support the study’s objectives. The protein array results were foundational, guiding the selection of candidate antigens for immunotherapy and informing subsequent experimental steps.

The primary aim of the study was to identify bacterial envelope proteins from Acinetobacter baumannii that could serve as promising targets for immunotherapeutic strategies, including vaccines or antibody-based treatments. RNA sequencing was employed to identify proteins highly expressed in human serum, and these findings were cross-referenced with a protein microarray probed using mouse serum. This approach enabled the selection of antigens for polyclonal antibody production and in vivo screening. In addition to the positive control, three novel candidates were identified. The study further investigated the mechanisms of immune activation associated with these candidates.

The methodology used to identify immunogenic components of A. baumannii is innovative, particularly the comparative analysis of protein expression across multiple clinical strains in human serum. The inclusion of both encapsulated and non-encapsulated strains as controls adds rigor to the experimental design. Furthermore, the study provides valuable insights into antibody binding loci, dosing efficacy in murine models, differential effectiveness in pulmonary versus bloodstream infections, and the immunostimulatory properties of the polyclonal antibodies. Collectively, these findings represent a comprehensive and meaningful advancement in the development of targeted therapies against A. baumannii.

Reviewer #3: This manuscript by Palethorpe et al describes the identification of A. baumanii proteins as protective antigens. A protein array was created containing antigens expressed by clinical isolates, identified by RNAseq and genome comparisons. Putative antigens were identified by screening sera from A. baumanii infected mice for recognition of antigens on the microarray.

A set of antigens was selected for further testing and produced as recombinant proteins in E. coli. Serum from rabbits immunized with the individual antigens was passively transferred into naïve mice to demonstrate protection against infection. Several protective antigens were identified in this study, which provides the basis for developing prophylactic or therapeutic vaccines against A. baumanii. Mechanistic studies showed the involvement of complement factors and neutrophils in bacterial clearance.

Major comments:

1. Was LPS removed from the protein preparations? If not, LPS contamination can affect immune responses in vivo.

2. Were rabbits immunized with the protein antigens alone, or with an adjuvant, to generate antibodies? This should be clarified.

3. The number of animals per group, the sex of the animals, and whether the experiments were repeated, should be included in the legends of each figure.

4. Figure 2A will be easier to read if purified antigens and bacterial lysates were run on the same gel next to each other. Label each lane with the name of the lysate, rather than referring to them as left and right lanes.

5. ELISA assays use no sera as the negative control. Inclusion of serum from a cytoplasmic protein, or unrelated antigen would be a better control.

6. Treatment of a group of mice with isotype control antibody along with anti-Ly6G is an important control for the depletion experiments.

7. It is important to test whether active immunization with purified proteins generates protective immune responses in mice.

8. The histograms in Figure 5 should be labeled to improve readability of the figure.

**Part II – Major Issues: Key Experiments Required for Acceptance**

Reviewer #1: • The initial RNA-Seq experiments in broth versus ex vivo serum are practically not discussed at all. A proper detailed analysis and interpretation of this data would be extremely useful to the field and likely expand the interest in the study overall.

• The three novel candidates are only explored through the realm of being protein antigens. It would be useful to describe what they are in more detail and their potential roles in the cell. Data to support a role for these candidates in fitness (perhaps in serum?) or pathogenicity would also be interesting and add to the story overall. As it stands there biological role is largely ignored despite transcriptomic data and subsequent ELISA data highlighting their importance.

Reviewer #2: None

Reviewer #3: Testing LPS content in the purified proteins.

Isotype control antibody treatment of mice as a control for neutrophil depletion.

Active immunization with the recombinant proteins to demonstrate protection.

**Part III – Minor Issues: Editorial and Data Presentation Modifications**

Reviewer #1: Throughout the study different strains are used for different assays, more rationale is needed each time a strain is dropped or included in each assay for example Line 217 why select these two strains here? And why for example were AB5075 and the corresponding wza mutant not included here?

• Line 200 Clarify what KL type Ab3879 is here. 2o Only needs to be clearly identified in each of the relevant figure legends.

• Line 208 rationale should be given for the differences particularly when independent of KL type. Needs more interpretation in the discussion also.

• Line 232 reasons for the lack of alignment should be briefly mentioned here and expanded upon in more detail in the discussion.

• Line 240 more detail needed here. Why was this strain selected when mucin needed to support colonisation? Citation(s) needed here.

• Line 310 previously described as ex vivo serum?

• The RNA-Seq data does not appear to have been deposited in an open access data for reviewer analysis.

• Would also recommend including a statement that they Thai strains are available to the wider research community upon request or (and I couldn’t see that this was the case) be deposited in a centralized collection.

Reviewer #2: While I have no major concerns, I recommend that the authors:

• Explain why they staed that potential dimers were formed for Ag5in line 185 and why only for Ag5. Are the immnoblots doen on Native-Page gels?

• Comment on the essentiality of the proteins that were tested.

• The authors could also expand the discussion to include a more detailed comparison between the human and mouse serum microarray results.

Additionally, the following questions merit consideration:

• Could the identified candidates be used in combinatorial immunotherapy?

• How might prophylactic treatment be administered prior to confirmed infection?

• What is the intended target population for such therapies?

Reviewer #3: (No Response)

PLOS authors have the option to publish the peer review history of their article (what does this mean? ). If published, this will include your full peer review and any attached files.

**Do you want your identity to be public for this peer review?** For information about this choice, including consent withdrawal, please see our Privacy Policy .

Reviewer #1: No

Reviewer #2: No

Reviewer #3: No

**Figure resubmission:**

**Reproducibility:**



---

## [Decision Letter · Decision Letter 1]

9 Jan 2026

PPATHOGENS-D-25-01927R1

Identification of multiple *Acinetobacter baumannii* protein antigens as targets for potential immunotherapies using a novel protein microarray screening approach

PLOS Pathogens

Dear Dr. Brown,

Thank you for submitting your manuscript to PLOS Pathogens. We appreciate your patience with delays in the review process over the holiday break. Your revised manuscript was evaluated by members of the editorial board and one of the previous reviewers. All believe that your study is interesting and important and that revisions addressed previously raised concerns. Only minor concerns remain to be addressed:

1. Report the amount of LPS present in the methods section as requested by Reviewer 3

2. Correct the typo in the Author Summary, should be "One important AMR bacterium"

3. Line 285 (marked): "polyclonal IgG antibodies"?

4. Line 515 (marked): Revise to more clearly indicate that the RNAseq datasets were previously published, e.g. "Transcriptome analyses by RNAseq were previously described"

Please submit your revised manuscript within 14 days. If you will need more time than this to complete your revisions, please reply to this message or contact the journal office at plospathogens@plos.org. Please include the following items when submitting your revised manuscript:

We look forward to receiving your revised manuscript.

Kind regards,

Lauren D. Palmer, PhD

Guest Editor

PLOS Pathogens

D. Scott Samuels

Section Editor

PLOS Pathogens

Sumita Bhaduri-McIntosh

Editor-in-Chief

PLOS Pathogens

orcid.org/0000-0003-2946-9497

Michael Malim

Editor-in-Chief

PLOS Pathogens

orcid.org/0000-0002-7699-2064

**Journal Requirements:**

**Reviewers' Comments:**

Reviewer's Responses to Questions

**Part I - Summary**

Reviewer #3: The authors have provided comprehensive responses to the reviewers' comments and added additional data to support the use of new antigens as vaccine targets.

Concern: while LPS was quantified in the protein preparations, the amount of LPS present was not reported. Please add this to the methods section.

**Part II – Major Issues: Key Experiments Required for Acceptance**

Reviewer #3: (No Response)

**Part III – Minor Issues: Editorial and Data Presentation Modifications**

Reviewer #3: (No Response)

PLOS authors have the option to publish the peer review history of their article (what does this mean? ). If published, this will include your full peer review and any attached files.

**Do you want your identity to be public for this peer review?** For information about this choice, including consent withdrawal, please see our Privacy Policy .

Reviewer #3: No

**Figure resubmission:**
---

## [Editor Report · Decision Letter 2]

30 Jan 2026

Dear Dr. Brown,

We are pleased to inform you that your manuscript 'Identification of multiple *Acinetobacter baumannii* protein antigens as targets for potential immunotherapies using a novel protein microarray screening approach' has been provisionally accepted for publication in PLOS Pathogens.

Best regards,

Lauren D. Palmer, PhD

Guest Editor

PLOS Pathogens

D. Scott Samuels

Section Editor

PLOS Pathogens

Sumita Bhaduri-McIntosh

Editor-in-Chief

PLOS Pathogens

orcid.org/0000-0003-2946-9497

Michael Malim

Editor-in-Chief

PLOS Pathogens

orcid.org/0000-0002-7699-2064
---

## [Editor Report · Acceptance letter]

Dear Dr. Brown,

We are delighted to inform you that your manuscript, "Identification of multiple *Acinetobacter baumannii* protein antigens as targets for potential immunotherapies using a novel protein microarray screening approach," has been formally accepted for publication in PLOS Pathogens.

Best regards,

Sumita Bhaduri-McIntosh

Editor-in-Chief

PLOS Pathogens

orcid.org/0000-0003-2946-9497

Michael Malim

Editor-in-Chief

PLOS Pathogens

orcid.org/0000-0002-7699-2064